# Restoring Axonal Organelle Motility and Regeneration in Cultured FUS-ALS Motoneurons through Magnetic Field Stimulation Suggests an Alternative Therapeutic Approach

**DOI:** 10.3390/cells12111502

**Published:** 2023-05-29

**Authors:** Wonphorn Kandhavivorn, Hannes Glaß, Thomas Herrmannsdörfer, Tobias M. Böckers, Marc Uhlarz, Jonas Gronemann, Richard H. W. Funk, Jens Pietzsch, Arun Pal, Andreas Hermann

**Affiliations:** 1Dresden High Magnetic Field Laboratory (HLD-EMFL), Helmholtz-Zentrum Dresden-Rossendorf, D-01328 Dresden, Germany; kwonphorn@gmail.com (W.K.); t.herrmannsdoerfer@hzdr.de (T.H.); m.uhlarz@hzdr.de (M.U.); j.gronemann@hzdr.de (J.G.);; 2Institute of Anatomy, Technische Universität Dresden, D-01307 Dresden, Germany; 3Division for Neurodegenerative Diseases, Department of Neurology, Technische Universität Dresden, D-01307 Dresden, Germany; hannes.glass@med.uni-rostock.de; 4Translational Neurodegeneration Section “Albrecht Kossel”, Department of Neurology, University Medical Center Rostock, University of Rostock, D-18147 Rostock, Germany; 5Institute of Anatomy and Cell Biology, University of Ulm, D-89081 Ulm, Germany; tobias.boeckers@uni-ulm.de; 6Deutsches Zentrum für Neurodegenerative Erkrankungen (DZNE) Ulm, D-89081 Ulm, Germany; 7Dresden International University, D-01067 Dresden, Germany; 8Department of Radiopharmaceutical and Chemical Biology, Institute of Radiopharmaceutical Cancer Research, Helmholtz-Zentrum Dresden-Rossendorf, D-01328 Dresden, Germany; 9Faculty of Chemistry and Food Chemistry, School of Science, Technische Universität Dresden, D-01069 Dresden, Germany; 10Deutsches Zentrum für Neurodegenerative Erkrankungen (DZNE) Rostock/Greifswald, D-18147 Rostock, Germany; 11Center for Transdisciplinary Neurosciences Rostock (CTNR), University Medical Center Rostock, University of Rostock, D-18147 Rostock, Germany

**Keywords:** amyotrophic lateral sclerosis, induced pluripotent stem cells, AC electromagnetic stimulation, axonopathy, axonal organelle trafficking, axonal sprouting, axonal regeneration, microtubules stabilization

## Abstract

Amyotrophic lateral sclerosis (ALS) is a devastating motoneuron disease characterized by sustained loss of neuromuscular junctions, degenerating corticospinal motoneurons and rapidly progressing muscle paralysis. Motoneurons have unique features, essentially a highly polarized, lengthy architecture of axons, posing a considerable challenge for maintaining long-range trafficking routes for organelles, cargo, mRNA and secretion with a high energy effort to serve crucial neuronal functions. Impaired intracellular pathways implicated in ALS pathology comprise RNA metabolism, cytoplasmic protein aggregation, cytoskeletal integrity for organelle trafficking and maintenance of mitochondrial morphology and function, cumulatively leading to neurodegeneration. Current drug treatments only have marginal effects on survival, thereby calling for alternative ALS therapies. Exposure to magnetic fields, e.g., transcranial magnetic stimulations (TMS) on the central nervous system (CNS), has been broadly explored over the past 20 years to investigate and improve physical and mental activities through stimulated excitability as well as neuronal plasticity. However, studies of magnetic treatments on the peripheral nervous system are still scarce. Thus, we investigated the therapeutic potential of low frequency alternating current magnetic fields on cultured spinal motoneurons derived from induced pluripotent stem cells of FUS-ALS patients and healthy persons. We report a remarkable restoration induced by magnetic stimulation on axonal trafficking of mitochondria and lysosomes and axonal regenerative sprouting after axotomy in FUS-ALS in vitro without obvious harmful effects on diseased and healthy neurons. These beneficial effects seem to derive from improved microtubule integrity. Thus, our study suggests the therapeutic potential of magnetic stimulations in ALS, which awaits further exploration and validation in future long-term in vivo studies.

## 1. Introduction

ALS is a severe incurable motoneuron disease disorder that rapidly progresses within typically two to five years until death. A progressive loss of lower (spinal) and upper (cortical) motoneurons causes paralysis, muscle atrophy and failure of the respiratory system at later stages [1]. Only 10% of ALS patients are familial cases (fALS) with positive anamnesis and/or respective inherited genetic mutations. Mainly missense point mutations are found in diverse genes such as Superoxide Dismutase 1 (SOD-1), Transactive Response Deoxyribonucleic acid (DNA) Binding Protein (TAR-DNA-binding protein-43; TDP-43), Fused in Sarcoma (FUS)/ translocation in liposarcoma, and GGGGCC hexanucleotide repeat expansion in Chromosome 9 Open Reading Frame 72 (C9ORF72) gene, the latter accounting with 40–50% for the majority of fALS cases [2]. However, most ALS patients are sporadic ALS (sALS) of yet unidentified causes, with no family record. While most ALS cases have their disease onset at the age of 60–70 years, some cases often bearing FUS or SENATAXIN mutations have their onset already in the late 20’s to 30’s [3,4,5,6]. The worldwide prevalence in all ages was 4.5 in 100,000 people with more rare cases before 50 and a peak incidence at 70 years of age [7]. Existing treatments are appallingly insufficient as they prolong patients’ lifespans by just a few months either with Riluzole, Edaravone or AMX00035, which are to date the only three available FDA—approved drugs [8,9,10]. Intracellular hallmarks of ALS pathology comprise toxic RNA accumulation, e.g., in TDP-43, C9ORF72 and FUS mutants, impaired protein quality control leading to, e.g., aberrant folding after translation, cytoskeletal disruption, perturbed axonal organelle trafficking and mitochondrial morphology and function [11]. Within the FUS gene, distinct mutations can occur in its diverse functional domains for transcriptional regulation, DNA recognition, DNA/RNA binding and the carboxyterminal nuclear localization signal (NLS). In particular, mutations in the NLS—by far the most common ones—are causative for protein aggregation, RNA incorporation into cytoplasmic stress granules, oxidative stress, and disruption of RNA and protein homeostasis [12]. Previously, we and others reported defects in mitochondrial and lysosomal trafficking toward distal axonal endings in spinal motoneurons derived from induced pluripotent stem cells (iPSCs) donated by FUS-ALS patients [13,14]. These cellular pathomechanisms gradually lead to neuromuscular junction dysfunction, followed by dying back [13,14,15].

Magnetic field stimulations realized with alternating (AC) or pulsed currents have been developed as non-invasive physical applications for neurological CNS diagnosis and therapy through inducing locally focused mild electrical currents in the conductive medium of brain tissue. In particular, TMS has been an established CNS application for psychiatric and neurophysiologic disorders for over two decades [16,17,18]. However, applications in peripheral nerves are scarce. Remarkably, studies of magnetic stimulation (MS) at 10 Hz for 10 min and then at 50 Hz for 10 min (20 min in total) every two days for 3 sessions in mice with traumatic injuries at muscles revealed an accelerated muscle and nerve regeneration through improved target finding of outgrowing axonal growth cones to new neuromuscular junctions [19]. Moreover, neuromuscular magnetic stimulation at 5 Hz of 50 stimuli (140 trains per stimulus with 15 s resting interval) for ten days (five days per week) was employed to improve muscle strength in ALS patients through augmented reinnervation at neuromuscular junctions [20]. To deepen insights into this approach, we explored the therapeutic potential of magnetic stimulation on cultured spinal motoneurons (MNs) derived from FUS-ALS patients and healthy test persons through iPSC technology. We measured axonal motility of mitochondria and lysosomes by live cell imaging since we previously documented the crucial role of the impaired trafficking of these organelles in neuronal survival and degeneration in various ALS mutants [13,21]. In addition, since the loss of neuronal connectivity at neuromuscular junctions in ALS is associated with a diminished ability of axons to outgrow and reconnect again, i.e., to regenerate [22]), we employed a refined assay to accurately analyze the new outgrowth of axonal growth cones after distal axotomy in compartmentalized microfluidic chamber cultures through live imaging.

## 2. Materials and Methods

### 2.1. Characteristics of Patients for iPSC Derivation

We studied iPSC-derived spinal motoneuron (MN) cell cultures from fALS patients bearing clinically mild FUS R521L^het^ and R521C^het^ and severe FUS R495QfsX527^het^ and P525L^het^ mutations and compared them to MNs carrying human wild type (WT) FUS alleles in three cell lines from healthy volunteers (WT, Ctrl1-3). Moreover, parental FUS R521C was used to generate isogenic FUS P525L-EGFP and its gene-corrected control FUS WT-EGFP [13]. All cell lines were obtained by skin biopsies of patients and healthy volunteers and have been described previously [23,24,25] (Table 1). The performed procedures were in accordance with the Declaration of Helsinki (WMA, 1964) and approved by the Ethical Committee of the Technische Universität Dresden, Germany (EK 393122012 and EK 45022009) and the Ethical Committee of the University of Ulm (Nr. 0148/2009). Written informed consent was obtained from all participants for the publication of any research results. Details of ALS patients and healthy donors were as follows:

### 2.2. Genotyping

DNA from the cell lines was genotyped by a diagnostic human genetic laboratory (CEGAT, Tübingen, Germany). Control lines were also genotyped and did not show any ALS mutation in SOD1, TDP43, FUS and C9ORF72.

### 2.3. Mycoplasma Testing

Every cell line was checked for mycoplasma when entering the lab, after reprogramming and routinely afterwards every three to six months. We used the Venor GeM Classic mycoplasma detection kit according to the manufacturer’s instructions (Cat No. 11-1025, Minerva Biolabs, Berlin, Germany).

### 2.4. Generating, Gene-Editing and Differentiation of Human iPSC Cell Lines to MNs in Microfluidic Chamber

The generation and expansion of iPSC lines from healthy controls and fALS patients with defined mutations in distinct ALS genes (Table 1) were previously described [13,23,24,25]. Briefly, fibroblast or keratinocyte lines were established from skin biopsies or hair follicle cells. These primary donor cells were reprogrammed using pMX-based retroviral expression vectors encoding the human cDNAs of *OCT4*, *SOX2*, *KF4* and *cMYC*. Transduced cells were further maintained on a mouse embryonic fibroblast feeder layer to generate colonies that were picked and repeatedly passaged to finally obtain iPSC clones. The obtained iPSC clones were further expanded and validated by germ layer differentiation (meso-, endo-, ectoderm), karyotyping, qRT-PCR to confirm the silencing of viral transgenes and finally genotyped to confirm the presence of the original donors’ FUS mutations or WT state, respectively. To generate the two isogenic lines, FUS WT-EGFP and FUS P525L-EGFP, the FUS R521C line was used as the parental source (Table 1). The patient-specific FUS R521C mutation was altered at its mutation site and simultaneously carboxyterminally tagged with EGFP by CRISPR/Cas9-mediated genome editing to obtain a P525L mutation, along with a gene-corrected WT control. Both new lines were fully characterized [13]. The subsequent differentiation of all iPSC lines to neuronal progenitor cells (NPC) and further maturation to spinal MNs was described previously [13,26]. In brief, colonies of iPSCs were collected and replated in human stem cell medium (hESC), containing 10 μM SB-431542, 1 μM Dorsomorphin, 3 μM CHIR 99021 and 0.5 μM Pumorphamine (PMA). After 2 days, hESC medium was replaced by N2B27 consisting of DMEM-F12/Neurobasal 50:50 supplemented with the aforementioned factors and 1:200 N2 Supplement, 1:100 B27 lacking Vitamin A and 1% penicillin/streptomycin/glutamine. On day 4, 150 μM Ascorbic acid was added, while Dorsomorphin and SB-431542 were withdrawn. Two days later, the obtained embryoid bodies were mechanically separated and replated on Matrigel-coated dishes. For this purpose, Matrigel was diluted (1:100) in DMEM-F12 and kept on the dishes overnight at room temperature. Possessing a ventralized and caudalized character, the arising so-called small molecule NPCs (smNPCs) formed homogenous colonies during the course of further cultivation. Growing colonies were split once per week at a ratio of 1:10–20 using Accutase for 5 min at 37 °C. For initiating the pre-differentiation at 0 DIV (Figure 1a), NPCs were maintained in pre-differentiation medium for motoneuronal patterning, which consisted of N2/B27 basal medium containing 1:1 of DMEM/F12:Neurobasal medium, 1% (*v*/*v*) L-Glutamine, 1% (*v*/*v*) 100 unit/mL Penicillin/100 µg/mL Streptomycin, 1% (*v*/*v*) B27 supplement and 0.5% (*v*/*v*), supplemented with 1 μM Retinoic acid, 0.5 μM SAG, 0.2 mM L-Ascorbic acid, 1 ng/mL BDNF and 1 ng/mL GDNF. At 6DIV, NPCs were supplemented with 5 ng/mL Activin-A into maturation medium for 24 h. The coating and assembly of microfluidic chambers (MFCs, Cat No. RD900, Xona Microfluidics, Research Triangle Park, NC, USA) to prepare for the seeding of MNs were performed as described [13,27]. MNs were seeded for maturation into one site of an MFC to obtain a fully compartmentalized culture with proximal somata and their dendrites physically separated from their distal axons, as only the latter type of neurite was capable of growing from the proximal seeding site through a microgroove barrier of 900 µm-long microchannels to the distal site (Figure 2a). The maturation medium was composed of N2/B27 basal medium supplemented with 0.1 mM bdcAMP, 0.2 mM L-Ascorbic acid, 2 ng/mL BDNF and 2 ng/mL GDNF and 1 ng/mL TGF β-3 that were added to the distal site of MFCs, whereas the medium added to the proximal site contained N2/B27 basal medium supplemented with only 0.1 mM bdcAMP and 0.2 mM L-Ascorbic acid in order to promote a chemotaxis towards distal compartments through the microchannels in MFCs for protruding axons. This compartmentalized culture format enabled the investigation of organelle trafficking, as well as the dynamics of growth cones specifically in axons aligned in parallel at defined anterograde versus retrograde orientation over the entire channel length. MNs matured for 30 days in vitro (DIV, D0 = day of starting differentiation, Figure 1a) prior to magnetic treatments and imaging.

### 2.5. Magnetic Field Stimulation (MS)

For our field stimulation experiments, we designed and fabricated electromagnetic coils (solenoids) with vertical cylindrical geometry that can be placed and operated in cell culture incubators. To compensate for heating beyond 37 °C due to ohmic losses, we equipped the coils with a water-cooling system. The solenoids with an open central bore of 50 mm diameter and a height of 100 mm could harbor several cell culture dishes (35 mm diameter and 12 mm height) within their central bore to expose them to a homogeneous field of perpendicular orientation to the axonal plane. For field generation, we applied alternating currents (AC) in square-wave form using a function generator connected to an amplifier. We would like to emphasize that the use of a rectangular excitation pattern results in an odd harmonic frequency spectrum of sine waves (e.g., 10 Hz rectangular waveform results in 10 Hz, 30 Hz, 50 Hz, 70 Hz, multi-harmonic frequencies and an associated amplitude distribution of 1, 1/3, 1/5, 1/7 with a converging decrease for odd higher harmonics). Up to six dishes with neuronal MFC samples were stacked within the upright central bore of each coil for simultaneous treatment. Each magnetic stimulation comprised four consecutive treatments (i.e., coil switched on) with intermittent resting intervals (i.e., coil switched off) and was performed with a biphasic rectangular waveform at a field strength of 10 millitesla (mT, time-average mean). These operational parameters were monitored with an oscilloscope to verify and monitor frequency and waveform and a Gauss meter to verify correct field strength. MS was performed with three distinct square-wave frequency settings: 2 Hz only or 10 Hz only in four repeated treatments or 10 Hz in three consecutive treatments and 2 Hz in the last treatment. Each treatment was continuously operated on for over 7 h, except the last treatment for only 3.5 h. The first treatment was performed at 30 DIV, the second at 35 DIV, the third at 40 DIV and the fourth at 45 DIV with field-less resting intervals for the cells between. The first treatment was performed when human MNs derived from FUS—ALS patients had already manifested phenotypic disease hallmarks, e.g., impaired axonal organelle trafficking [13]. After the fourth or final treatment, the cells were maintained in culture for two more resting days before the sustained impact of the MS was assessed at 48 DIV by live cell imaging and immunofluorescent staining (see below).

### 2.6. Immunofluorescent Stainings

Immunofluorescent stainings were performed according to standard protocols [13]. In brief, samples were fixed with 4% paraformaldehyde, permeabilized with 0.2% Triton X100 and blocked in a mix of 1% BSA and 5% goat serum. Primary antibodies were then employed as follows: anti-microtubule associated protein 2 (MAP2) (1:2000, Cat No. ab5392, Abcam) and anti-β3-tubulin (alias TUJ1) (1:500, Cat No. T5076, Sigma Aldrich, St. Louis, MO, USA) served as general neuronal markers and anti-neurofilament heavy chain (1:300, Cat No. smi-32P, BioLegend, San Diego, CA, USA) to detect heavy neurofilament H as MN marker (SMI32), anti-acetylated tubulin (1:400, Cat No. T7451, Sigma Aldrich, St. Louis, MO, USA) and anti-α tubulin (1:1000, Cat No. ab4074, Abcam, Cambridge, UK) to elucidate post-translational modification of β3-tubulin and α-tubulin, respectively, and anti-cleaved-caspase-3 (1:500, Cat No. ab32042, Abcam, Cambridge, UK) to determine apoptosis. The nuclei were counterstained with Hoechst 33342 (Cat No. 14533, Sigma Aldrich, St. Louis, MO, USA).

### 2.7. Immunofluorescence Intensity Analysis

To quantify the fluorescent intensity of neuronal and motoneuronal markers and others (see above), their mean intensities were analyzed in FIJI (Image J 1.53t) using standard thresholding and masking tools. Specifically, the percentage of marker-positive cells in Figure 1, Appendix A was determined by manual counting of marker-positive cells after thresholding using the respective plugin. The count was then normalized by the total cell count in the respective images, as determined with Hoechst33342. To measure cleaved caspase-3 percentages (CC3), a binary mask was generated after thresholding the MAP2 channel and applied to the CC3 channel to restrict the analysis to neuronal cells only. Within this MAP2 mask, the percentage of CC3-positive neurons was determined, as shown in Figure 1. To measure MAP2 and SMI32 intensities, a background correction was performed using a blank area between neurites followed by normalization by total neurite length using the skeletonize tool. Settings for thresholds and background corrections were equal across all conditions (genotype, MS frequency, etc.) to eliminate any bias. To measure the level of acetylated tubulin, a background correction was performed using a blank area between the neurites for both the acetylated and α-tubulin channel. Next, the intensity of acetylated tubulin was normalized by α-tubulin. The results obtained for all figures were on a per-image base. Images were acquired with the same technical setting in all microscope sessions across all conditions (genotype, MS frequency, etc.) to eliminate any bias, e.g., with respect to laser power, exposure time, amplification gain, etc. At least 3 independent experiments on distinct differentiation pipelines of each cell line were performed. Typically, 10 images per experiment for each cell line and condition were analyzed, as described above.

### 2.8. Axonal Live Cell Imaging of MNs in MFCs

To investigate axonal organelle motility, Ctrl and FUS MNs (Table 1) were matured in MFCs for 21 days (30 DIV after the start of differentiation, Figure 1a) [13] and MS at different frequencies was performed as described above. Time-lapse movie acquisition was performed as described [13,27]. In brief, to track lysosomes and mitochondria, cells were double-stained with live cell dyes Lysotracker Red DND-99 (Molecular Probes Cat. No. L-7528) and Mitotracker Deep Red FM (Molecular Probes Cat. No. M22426) at a final 50 nM each. Trackers were added from a 1 mM stock in DMSO directly to culture supernatants in both the proximal and distal compartments of MFCs and incubated for 1 h at 37 °C. Live imaging was then performed in the Center for Molecular and Cellular Bioengineering, Technische Universität Dresden (CMCB) light microscopy facility with a Leica HC PL APO 100 × 1.46 N/A oil immersion objective on an inversed fluorescent Leica DMI6000 microscope enclosed in an incubator chamber (37 °C, 5% CO_2_, humidified air) and fitted to a 12—bit Andor iXON 897 EMCCD camera (521 × 512 pixel, 16 µm/ pixels on chip, 229.55 nm/pixel at 100× magnification with intermediate 0.7× demagnification in the optical path through the C-mount adapter connecting the camera with the microscope). For more details, refer to https://www.biodip.de/wiki/Bioz06_-_Leica_AFLX6000_TIRF (8 March 2023). Fast dual color movies were recorded at 3.3 frames per second (fps) per channel over 2 min (400 frames per channel in total) with 115 ms exposure time as follows: Lysotracker Red (excitation: 561 nm Laser line, emission filter TRITC 605/65 nm) and Mitotracker Deep Red (excitation: 633 nm Laser line, emission filter Cy5 720/60). Dual channel imaging was achieved sequentially by fast switching between both laser lines and emission filters using a motorized filter wheel to eliminate any crosstalk between both trackers. At least 5–6 movies were acquired from each readout position per MFC, resulting in at least 15 movies per condition (genotype, MS frequency, readout position, etc.) per experiment due to 3 technical replicas (=MFCs). At least three independent experiments on independent differentiation pipelines were performed, yielding at least 45 movies per condition (genotype, MS frequency, readout position, etc.). Each movie covered 2 MFC microchannels in its viewing field that were typically populated by 5–30 axons, thereby resulting in hundreds of organelles recorded and tracked (see below) per movie.

### 2.9. Axonal Organelle Tracking and Shape Analysis of Live Imaging Movies

A comprehensive description of the automated analytical pipeline, starting from object recognition in raw movie data to final parametrization of organelle motility and morphology, was previously described [27]. In brief, the analysis yielded bulk statistics with data values on a per-organelle base without assigning individual organelles to a particular axon within a microchannel due to the technical limitations of the microscope (i.e., optical resolution) and because such an assignment was irrelevant for the sought-after analysis. Moreover, different axon numbers across the channels did not introduce an error per se; inter-channel variability was averaged due to the subsequent pooling of thousands of tracks across many channels and movies (see below). In essence, we never observed that the organelle motility in a bundle composed of, e.g., 10 axons is different from a bundle composed of 20 axons. Any inter-channel variability was just part of the overall variability in the whole pooled organelle population. Organelle recognition and tracking with Mito- and Lysotracker was performed with the FIJI (Image J 1.53t) TrackMate plugin and organelle shape analysis with our custom-tailored FIJI Morphology macro, which is based on the FIJI (Image J 1.53t) particle analyzer. TrackMate returned a multitude of dynamic tracking parameters, i.e., numerical descriptors, for a comprehensive phenotypic profiling of each track through multiple parameterization. Examples of these dynamic tracking parameters comprised organelle mean speed and track displacement; the latter served as a measure for straight, processive movement as opposed to a non-directional random walk. A full list of the obtained parameters is provided below. Likewise, the Morphology macro returned static shape parameters of either type of organelle. Specifically, the diameter of lysosomes was measured with its outer Feret diameter, i.e., the outer circumference fitted to globular objects, whereas the elongation of mitochondria was expressed through its aspect ratio, i.e., the ratio of major over minor radius of its fitted eclipse. Subsequent data mining of individual per-movie result files was performed in KNIME 4.5.3 to assemble final result files with annotated per-organelle parameters, thereby allowing all data from each experimental condition to be pooled (e.g., all data for Mito- or Lysotracker for a given cell line treated at a particular frequency). Data per organelle were visualized as box plots for mean speed and bar graphs for percent moving tracks in Figure 2.

### 2.10. Multiparametric High Content (HC) Phenotypic Profiles

The FIJI (Image J 1.53t)Track Mate and Morphology plugins returned a multitude of dynamic organelle tracking as well as static shape parameters (see above), of which the following 11 were further processed for HC phenotypic profiling: # 1; track duration, # 2; track displacement, # 3; mean speed, # 4; max speed, # 5; min speed, # 6; median speed, # 7; standard deviation (SD) of speed, # 8; track length; # 9; percent moving track; # 10; ratio anterograde/retrograde tracks and # 11; either elongation of mitochondria or diameter of lysosomes. This set of 11 master parameters was deduced over the entire batch organelle population of each condition (i.e., movie pool of each condition, e.g., Lysotracker\Distal\Treatment\Cell lines\all movie) and the normalized deviation from Ctrl untreated at the proximal readout (i.e., the reference baseline), expressed as a Z-score as previously described [27]. The Z-scores of each parameter were connected with a line to yield the shown profiles for better clarity in revealing significant changes for individual parameters. The normalization to Ctrl untreated parameters at the proximal site resulted in a flat line for the HC profile of Ctrl untreated for its proximal parts due to normalization to itself (i.e., Z-score = 0), whereas its distal parts mirrored the deviations of distal organelle motility from their proximal counterparts even under physiological conditions. Thus, the untreated Ctrl profiles in their distal parts are not flat lines. The entire resultant profiles comprised the above master set of 11 parameter-scores four times, namely for Mitotracker at the distal MFC channel site (parameter # 1–11) versus Mitotracker at the proximal site (# 12–22) and ditto for distal Lysotracker (# 23–33) versus proximal Lysotracker (# 34–44), i.e., 44 parameters in total.

### 2.11. Live Cell Imaging of Growth Cones

To mimic and model peripheral nerve injuries, the distal axons of MNs cultured in MFCs were mechanically axotomized at 48 DIV, as described [28,29,30]. In brief, shearing forces were applied through pipette action at the distal MFC reservoirs, i.e., ~200 µL of medium were forcedly triturated through the corridor of the distal MFC compartment. Damage of all axons was verified under a microscope and if it was incomplete, the procedure was repeated with fresh medium until an entirely blank assay area at the distal channels’ exit was established (Figure 4a and Appendix A). At the end of the axotomy, axotomized MNs were replaced with fresh medium and placed back into the incubator for 24 h, at which point the cells were imaged. Image acquisition was performed on all Ctrl and FUS lines (Table 1) with a Zeiss 40×/1.3 Oil DICIII objective on an inverted Zeiss AXIO Observer Z1 microscope stand equipped with a sCMOS camera (OptiMOS™) and an incubator chamber (37 °C, 5% CO_2_ and humid air). At least four DIC bright field movies were acquired per MFC at 3 frames per min over 2 h. Movies were cropped down to individual growth cones for further analysis of outgrowth speed using a custom-tailored macro in FIJI (see below). At least 10–20 growth cones were analyzed per line in each experiment. At least 3–4 independent experiments on distinct differentiation pipelines of each cell line were performed.

### 2.12. Outgrowth Analysis of Advancing Growth Cone Velocity

Movies of newly outgrowing growth cones obtained after distal axotomy (see Section 2.11.) were analyzed using a custom-tailored macro for FIJI (Image J 1.53t). The macro detected the individual growth cones in x-y coordinates from each frame of the movie stack and subsequently calculated the mean travel distance of the growth cones between consecutive frames to deduce their mean outgrowth velocity. Each movie was cropped down to regions, each covering the entire migration of a single growth cone. The analysis was performed stepwise, as follows:In FIJI, the macro “Growth cone characteristics” was loaded and executed. User input was then required to locate the parental folder containing all movie stacks.The pixel dimension calibrations from the microscope were automatically imported from the movie’s metadata and implemented for the analysis.For each movie, a rectangular ROI for the initial growth cone detection was manually drawn, which covered the entire migration throughout the whole movie stack. The ROI was carefully selected to eliminate erratic detections of objects other than growth cones. Some preliminary image optimizations were conducted in the ROI, as follows:
◦Contrast enhancement by allowing 0.1% saturated pixels and histogram normalization,◦Background subtraction with an eroding rolling ball of 20 pixels in radius on light background,◦Image segmentation with the thresholding function “Percentile” on a dark background,◦Background setting to black,◦Conversion of the obtained segmented images to masks.

Hereafter, the growth cones were clearly identified in each ROI.
Growth cone tracking was performed on the black/white masks of all segmented objects. Given the dynamic nature of the growth cone’s morphology over consecutive frames, their tracking required a new determination of their center of mass for each frame. This was achieved by an iterative mask shrinking process starting from the borders of the selected ROI via decreasing circular masks following an intensity gradient that finally shrunk down to the intensity center of each recognized object to determine its current coordinates, thereby enabling its linking to consecutive frames. As a pragmatic approximation, it was assumed that the center of intensity equated to the center of mass. The radius was successively reduced until some selection remained. The remaining selection was then re-inflated back to its original size and the center-of-mass was calculated and highlighted in detected growth cones.In the case of several growth cones in the ROI, the user had to select a single growth cone of interest for further analysis.The selected growth cone (outlined in Figure 3b) was automatically analyzed by the macro with respect to its mean travel distance between consecutive frames over the entire movie stack (Figure 3b). To this end, the algorithm scanned an area surrounding the x-y coordinates of the cone’s center of mass deduced in the previous frame. In the case of multiple plausible positions, the algorithm chose the object with the closest distance to the previous frame as the new position. This process was repeated frame by frame.

The full macro code is provided in Appendix A. In essence, our macro was faithfully locating and linking each growth cone in all consecutive frames within the ROIs and reliably coping with their dynamic morphological fluctuations. The x-y coordinates of each growth cone over the frame number were saved in individual per-movie CSV result files. Further data mining, calculations of mean speed as well as distance and final result assembly were performed in KNIME 4.5.3.

### 2.13. Quantification and Statistics

All experiments were analyzed by the same experimenter using strictly standardized, pre-defined readout positions and settings, followed by semi-automatic image analysis. Even not blinded (since the intracellular FUS-GFP pattern is clearly different in mutant FUS versus WT FUS, thereby deblinding the genotype automatically), these standardized procedures ensured the minimization of any bias in the quantification procedure. Moreover, we were at least blinded to the treatment condition (untreated versus MS). Images were acquired at random positions in cell culture dishes at 48–50 DIV after having completed all MS except for the standardized readout positions at distal versus proximal channel positions in MFCs. At least three experiments were performed on three independent MN differentiation pipelines. Statistical analysis was performed using Origin software. Normal distributions of data sets were tested using the Kolmogorov–Smirnov method. One-way ANOVA with Bonferroni or otherwise the Kruskal–Wallis post-hoc test or Student’s *t*-test was utilized to reveal significance differences in pairwise comparisons. Data values are shown as means ± standard deviation (SD), unless otherwise indicated. The Bonferroni post-hoc test on two-way ANOVA was employed to reveal significant differences between conditions in multiple pairwise comparisons on averaged Z-scores of multiparametric HC profiles. Asterisks: highly significant alteration in indicated pairwise comparisons, * *p* ≤ 0.05, ** *p* ≤ 0.01, *** *p* ≤ 0.001 and **** *p* ≤ 0.0001, ns: no significant difference.

## 3. Results

### 3.1. Neuronal Characterization and the Effect of Magnetic Stimulations on Neuronal Differentiation

We performed a preliminary basal characterization of our iPSC-derived spinal motoneurons to determine whether their differentiation was altered by FUS mutations or magnetic stimulation. Collectively, four FUS mutant lines, i.e., the P525L and R521C isogenic and R521L and 495QfsX527 non-isogenic lines along with four healthy control lines, i.e., three lines from three healthy probands and the wild-type (WT) isogenic to the P525L and R521C mutant lines, were utilized in this study (Table 1) and characterized by immunofluorescent stainings for, first, the general neuronal marker microtubule-associated protein 2 (MAP2) and, second, the motoneuron-specific marker neurofilament heavy chain H (alias SMI32) to assess the differentiation process (Figure 1b and Appendix A).

The mean percentage of marker-positive cells was calculated by normalization with the total number of nuclei in each image. We revealed a mean percentage of MAP2-positive neurons ranging from 60–80%, with no significant differences across all mutant FUS and Ctrl lines, thereby validating the data pooling of all mutant FUS and Ctrl lines (Figure 1). For results on individual lines, refer to Appendix A. Furthermore, we revealed no difference between the Ctrl and mutant FUS line pools (Figure 1c). The fraction of cells positive for SMI32 was indistinguishable for the Ctrl versus mutant FUS pool (Figure 1d), irrespective of whether the calculation was done with respect to the entire Hoechst-positive cell population (Figure 1d) or was limited to the neuronal MAP2-positive population (Figure 1e). Collectively, our basal neuronal characterizations revealed no obvious differences in the capacity of our iPSC lines to differentiate to MNs irrespective of genetic background and mutation, consistent with our previous report [13].

In a previous study, electromagnetic stimulation of murine NPCs with 1 mT at 50 Hz led to an increased percentage of MAP2- and TUJ1-positive cells [31]. This previous study was performed on primary neural stem /progenitor cells (NSCs) isolated from brain cortices of newborn mice. Thus, we sought to investigate whether our magnetic stimulation protocol impacted on neuronal differentiation in response to repeated MS at 10 Hz versus untreated controls (detailed in Section 2.5., Material and Methods). The mean percentages of MAP2-positive Ctrl cells (pooled Ctrl1 and FUS-GFP WT, Table 1) versus mutant FUS (pooled FUS R495QfsX527 and FUS-GFP P525L) were 70.18 ± 6.43 and 69.86 ± 2.22 for untreated cells and 69.46 ± 5.78 and 65.59 ± 1.23 for MS, respectively (Figure 1f). For the overall motoneuron subtype specification, the mean percentages of SMI32-positive Ctrl versus mutant FUS cells were 55.78 ± 6.46 and 53.86 ± 2.73 for untreated cells and 57.26 ± 7.01 and 53.05 ± 3.90 for MS, respectively (Figure 1g). The number of MNs (percentage of SMI32-positive cells within the MAP2-positive population; % SMI32^+^/MAP2^+^ neurons) in the Ctrl versus mutant FUS pool did not differ and was 78.5 ± 4.26 and 76.32 ± 6.42 for untreated cells and 81.20 ± 3.82 and 74.82 ± 6.63 for MS, respectively (Figure 1h). In essence, magnetic stimulations had no obvious impact on neuronal and motoneuronal differentiation and maturation on any individual line (Appendix A) or on the Ctrl and mutant FUS line pool (Figure 1), consistent with the late time point of MS at 48 DIV (Figure 1a), when MNs were already in advanced maturation [13].

### 3.2. MS Restores Deficient Axonal Organelle Transport in FUS-ALS

The highly polarized, lengthy architecture of neurites, particularly axons, poses a considerable challenge in maintaining long-range trafficking routes that serve crucial neuronal maintenance and functions. Axonal organelle transport plays a crucial role in neuronal activities, e.g., terminal synaptic communication and regeneration [32]. Thus, impairment of axonal organelle motility can contribute to neurodegeneration in, e.g., ALS and Parkinson’s disease [33]. We previously reported impairments in axonal organelle motility caused by mutations in the FUS gene in spinal MNs derived from fALS patients [13]. To investigate any beneficial impact of MS on organelle motility in axons, MNs were differentiated and maturated in compartmentalized Zona microfluidic chambers (MFCs), in which the soma-dendritic seeding compartment is physically separated from the axonal distal compartment by a microgroove barrier of microchannels (Figure 2a) [13,27]. Only axons but not dendrites [34], are capable of fully growing through the microchannels and sprouting out at the distal exit, thereby enabling axon-specific imaging at the distal versus the proximal channel end with defined bundle alignment in parallel and directionality antero- versus retrograde trafficking events. We performed live imaging with Mito- and Lysotracker at strictly standardized readout positions at the distal versus proximal channel ends to reveal any phenotypic changes with increasing distance to somata, as described [13,27]. Maximum intensity projections of entire movie tracks enabled a preliminary visual inspection of major alterations in motility patterns (Figure 2b and Corresponding Appendix A). Direct, processive trafficking events were highlighted as long trajectories, whereas stationary organelles and non-processive “jitter” remained virtually as punctae. These maximum intensity projections revealed an apparent reduction of mitochondrial and lysosomal organelle motility in untreated mutant FUS MNs at the distal channel end compared to Ctrl cells (Figure 2b, Appendix A), whereas proximal motility remained physiological, consistent with our previous report [13]. In our search for a meaningful frequency setting for our MS experiments, we encountered a growing number of published reports about the beneficial effects of extremely low frequency electromagnetic field (ELF-EMF) applications in the range of 1–100 Hz [19,20,31]. In essence, it was difficult for us to decide which of the already tested frequencies could serve as the most promising and meaningful starting point for our own MS experiments on human iPSC-derived MNs. This was simply due to the multitude of different disease models and biological readout assays tested with unclear relevance for our own model system and the biological questions we wished to address [19,20,31]. Thus, in order to streamline our proof of concept study and to avoid tedious screenings to empirically establish efficient frequency settings, we decided to use AC rectangular instead of sine waves. Unlike homogenous sine waves, rectangular waves have the benefit of producing a spectrum of higher harmonic frequencies according to Fourier analysis (see also Section 2.5., Material and Methods), thereby allowing multiple frequencies to be tested simultaneously. For example, a basal frequency of 2 Hz rectangular waves results in 2 Hz of amplitude 1 + 6 Hz of amplitude 1/3 + 10 Hz of amplitude 1/5 + 14 Hz of amplitude 1/7 + etc. Likewise, a basal frequency of 10 Hz results in 10 Hz of amplitude 1 + 30 Hz of amplitude 1/3 + 50 Hz of amplitude 1/5 + 70 Hz of amplitude 1/7 + etc. We decided to test 2 Hz to capture the low ELF-EMF range and 10 Hz to capture the higher range. To counterbalance the decreasing amplitude at higher harmonic frequencies, we utilized a relatively high field strength of 10 mT, which was considerably higher than in many published reports (e.g., in [31]: 1 mT at 50 Hz). We also sought to combine both overlapping harmonic ranges by sequentially exposing our MNs first to 10 Hz, followed by 2 Hz. To this end, we established in preliminary trials that a sequence of three 10 Hz treatments followed by one final 2 Hz treatment (see Section 2.5. Material and Methods) was optimal in combining the beneficial effects of both settings. Therefore, we included the combinatorial 10/2 Hz treatment in the following assays.

Indeed, MS led to a restoration of distal motility for either type of organelle, whereas no further boost or other alteration through MS was visible in Ctrl MNs (Figure 2b, Appendix A). Consistently, our established quantitative tracking analysis [13,27] revealed a decreased distal mean speed in box plots (for per-organelle values) and percentage of moving tracks in bar graphs (means per-movie values) for mitochondria (Figure 2c,d) and lysosomes in untreated mutant FUS MNs as compared to Ctrl cells (Figure 2e,f), respectively (compare untreated Ctrl in blue versus untreated mutant FUS in red at the distal site in Figure 2c–f). These distal defects in mutant FUS reverted back to Ctrl levels through MS, depending on the AC frequency used. Specifically, the distal mitochondrial mean speed in mutant FUS was best rescued at combinatorial 10 and 2 Hz (10/2 Hz), to a lesser extent at 10 Hz only as well, but with no alteration at 2 Hz only (Figure 2c, compare distal red box plots “untreated” versus “2 Hz”, “10 Hz”, and “10/2 Hz”). We revealed a similar result for the distal lysosomal mean speed in mutant FUS with the most appreciable rescue at combinatorial 10/2 Hz, whereas only 2 and 10 Hz were far less efficient (Figure 2e, compare distal red box plots “untreated” versus “2 Hz”, “10 Hz”, and “10/2 Hz”). Of note, the physiological distal Ctrl levels remained unaltered (Figure 2d,f, compare distal blue box plots “untreated” versus “2 Hz”, “10 Hz”, and “10/2 Hz”). As for the proximal mobile fraction, we did not see any deterioration through our MS protocol at any frequency, either in mutant FUS or in Ctrl MNs, but some further increase in controls (Figure 2d,f, proximal box plots), consistent with the apparently unaltered organelle motility in mutant FUS as compared to Ctrl MNs in the corresponding raw data (Figure 2b proximal).

Next, we wished to analyze the impact of MS on these complex axonal trafficking logistics in more detail, as a tracking analysis restricted to mean speed only might be insufficient to understand the extent to which MS was capable of restoring a pathological impairment back to phenotypic wild-type levels. To this end, we took advantage of our established phenotypic high content (HC) profiling method for a more comprehensive analysis of axonal organelle motility [21,27]. In brief, we deduced multiparametric HC profiles for each line (Table 1) that contained the aforementioned mean speed and track displacement along with other numerical descriptors to obtain a master set of 11 parameters (detailed in Materials and Methods) four times, owing to two readout positions (distal versus proximal) and two markers (Mito- and Lysotracker) assembled to a signature of 44 parameters in total. Each parameter was expressed as a Z-score deviation from pooled untreated WT control lines at the proximal readout and plotted with a connecting line to obtain a graphical signature, i.e., the HC profile (Figure 3a–c). In essence, the multiparametric HC profile served as a quantitative, comprehensive readout to assess natural disease states and their restorations back to wild-type levels through MS. We tested the impact of the AC field at 2 Hz (Figure 3a), 10 Hz (Figure 3b) and the combinatorial sequence of 10/2 Hz (Figure 3c) (detailed in Materials and Methods). Due to moderate inter-line variability, each parameter Z-score was averaged over all four control and three mutant FUS lines, respectively, to obtain a consensus mean profile for the Ctrl (in blue) and mutant FUS pools (in red), with error bars representing the deviation across the individual lines (Figure 3a–c). The HC profiles of individual cell lines are provided in Appendix A.

The HC profiles of untreated Ctrl MNs exhibited milder deviations compared to the mutant FUS pool, as expected. However, some more prominent negative deviations occurred distally, particularly for lysosomes, e.g., in mean, max, min, median speed and its SD, parameter # 26–29, respectively, suggesting a natural physiological trafficking gradient for lysosomes from proximal to distal in axons (Figure 3a–c) due to aging, as we previously documented [27]. Conversely, the untreated mutant FUS MNs exhibited remarkable negative deviations in multiple tracking parameters for either type of organelles (mitochondria and lysosomes) at both readout sites (distal and proximal) (Figure 3a–c), particularly at distal sites in the mean, max, min median speed and its SD, parameter # 3–8 and 25–29, percent moving tracks (parameter # 10) and elongation of mitochondria (parameter # 11) and at the proximal sites only in mean, max, min, median speed and its SD, parameter # 14–18 for mitochondria and 36–37 and 39–40 for lysosomes. Even though some error bars were substantial, the negative deviation of multiple parameters in concert with respect to the Ctrl pool indicated the significance of this result, confirmed by two-way ANOVA on the untreated Ctrl pool versus the untreated mutant FUS pool for the entire profile (*p* ˂ 0.0001, Table 2).

As for MS, treatment at 2 Hz only led to no appreciable alteration of the entire HC profile, neither in the Ctrl, nor in the mutant FUS pool (Figure 3c, Table 2). Conversely, higher MS frequencies (10 Hz only and sequential 10/2 Hz) resulted in an overall beneficial reversion of the entire mutant FUS HC profile toward the Ctrl pool. Due to the higher comprehensive and conclusive power of the multiparametric HC analysis (Figure 3a–c) over the single parameter analysis (Figure 2c–f), we did not further pursue the difference seen between the 10 Hz only versus the sequential 10/2 Hz treatment seen in the latter. Instead, we concluded from the multiparametric HC analysis (Figure 3a–c, Table 2) that both the 10 Hz only and sequential 10/2 Hz MS treatments exhibited similar performance in rescuing axonal trafficking defects in mutant FUS MNs due to the higher frequency part that mattered, and thus, we conducted all other experiments with 10 Hz only treatments. Taken together, our HC profiling of axonal organelle motility further substantiated the beneficial MS effects on multiple organelle trafficking parameters. The higher efficiency at 10 Hz over 2 Hz indicates that the frequency of the alternating current matters. Finally, some milder stimulation beyond their physiological level was achieved in the Ctrl MNs, as well as through sequential 10/2 Hz treatments, thereby pointing to a generic beneficial boosting effect through magnetic stimulations, regardless of specific pathological states.

### 3.3. Rescue of Axonal Regeneration Defects in FUS-ALS through MS

Neuronal connectivity across nerve endings is crucial in both the CNS and PNS for transmitting information chemically and electrically in maintaining their functions, i.e., learning, memory, sensory, limb and organ control [35]. Maintaining this crucial inter-neuronal connectivity intimately relies on the ability of growth cones to regenerate after injuries or during aging and to grow out again for new target finding, i.e., for both structural and synaptic plasticity [36,37]. Likewise, the ability of peripheral MNs to control muscle contraction critically relies on properly maintained axonal connectivity across neuromuscular junctions (neuromuscular unit) and its regeneration or sprouting in response to damage and aging insults of the neuromuscular unit. The retarded regeneration of neuromuscular junctions leads to the denervation of muscles [38]. Impaired branching of neurons and distribution of axons were documented in mouse cortical neurons with FUS mutations [39]. Thus, we wished to investigate whether our untreated cellular mutant FUS models exhibited any regenerative outgrowth defects in their axons and to what extent our MS protocol was capable of improving such defects at 2 Hz only, 10 Hz only and sequential 10 Hz and 2 Hz, respectively. To this end, we chose a late end point at 48 DIV to better match the adult onset of ALS in vivo. At this time point, MNs were already progressively matured, with many axons having sprouted out from the distal channel end in MFCs [13]. We performed the previously reported method of mechanical axotomy through pipette action [28,29,30] to establish a blank assay area for regeneration at the distal MFC channel exits, which served as a truncation boundary (Figure 4a). We carefully verified that the assay area after the axotomy was devoid of any axons or debris (Appendix A). Moreover, we also verified that the remaining axon trunks after axotomy inside the MFC channels reached close to the distal channel exit, consistent with a sharp truncation boundary (Appendix A). After axotomy, we allowed the newly grown axons to repopulate the blank distal assay area for 24 h (Figure 4a,b) to then analyze the new growth cones by live cell imaging. Acquired movies were analyzed with a custom-tailored macro in FIJI to semi-automatically recognize and track the advancing growth cones and deduce their mean speed after MS versus untreated controls (detailed in Section 2.12. Materials and Methods, Figure 4b).

Indeed, we revealed a reduced mean outgrowth speed for growth cones in untreated mutant FUS compared to Ctrl MNs, namely 0.0530 ± 0.0210 μm/s (mean ± SD) in the mutant FUS pool versus 0.0666 ± 0.0224 μm/s (*p* ≤ 0.0001) in the Ctrl pool, as shown Figure 4c. After MS at 2 Hz only, 10 Hz only, and sequential 10/2 Hz, the mean outgrowth speed in Ctrl MNs remained unaltered at 0.0639 ± 0.0236 μm/s, 0.0696 ± 0.0255 μm/s and 0.0690 ± 0.0250 μm/s, respectively (Figure 4c). Conversely, MS on mutant FUS MNs at 10 Hz significantly increased the mean outgrowth speed to 0.0616 ± 0.0242 μm/s (*p* ≤ 0.05) consistent with a faithful reversion back to Ctrl levels, whereas 2 Hz only and sequential 10/2 Hz did not show significant changes. Taken together, these findings are in concordance with the beneficial effects of MS, especially at higher frequencies, on axonal organelle trafficking (Figure 2f and Figure 3a–c). As for individual lines, refer to Appendix A. Again, we found no obvious evidence of adverse side effects through MS on axonal regeneration, either on Ctrl or on mutant FUS MNs, such as retarded outgrowth or crippled cone morphology, thus pointing to a benign character of the MS treatment method. The apparently similar axon densities 24 h after axonotomy, regardless of genotype and treatment (Figure 4b), were not systematically investigated in this setup since we had to select for traceable growth cones, which might have biased such an analysis.

### 3.4. MS Did Not Alter Neuronal Survival

Next, we wished to rule out possible detrimental side effects and probe for signs of apoptosis after the execution of our MS protocol at 10 Hz on cultured MNs. To this end, we performed immunofluorescent staining for the apoptotic marker cleaved caspase-3 (Figure 5a,b) of one non-isogenic (Ctrl 1 versus the severe mutant FUS 495QfsX527, Table 1) and one isogenic pair (FUS-GFP WT versus FUS-GFP P525L, Table 1). We determined the apoptotic fractions (i.e., percentages) in untreated Ctrl and mutant FUS cultures in MAP2-positive neurons only, thereby capturing MNs and possibly other neuronal subtypes in our differentiation protocol. We found significantly augmented apoptosis in the untreated mutant FUS pool over the Ctrl pool (Figure 5c and Appendix A). These findings at 48 DIV (Figure 1a) are in good agreement with our previous report showing augmented apoptosis in the FUS mutants at 60 DIV and beyond [13]. When we tested our MS protocol, the apoptotic fractions did not change either in the Ctrl pool or in the mutant FUS pool (Figure 5c). However, there was a rescue in some individual lines, whereas we did not detect a harmful effect of MS in any cell line investigated (Appendix A).

### 3.5. Magnetic Stimulations Modulated Cytoskeleton Integrity in MNs with FUS Mutations

MAP2 plays a documented key role in microtubule stabilization and thus in nerve regeneration [40]. Likewise, tubulin acetylation is important for stabilizing microtubules from fragmentation, as a reduction of this post-translation modification caused microtubule instability [41,42]. Therefore, given that neuronal differentiation and maturation were unaltered in mutant FUS without or after magnetic treatments (Figure 1), we next addressed the possibility of compromised cytoskeletal integrity arising in later stages during neuronal aging as an underlying cause of ALS pathology. To this end, we first revealed MAP2 levels by IF microscopy in our matured Ctrl and mutant FUS MNs in untreated versus treated (MS at 10 Hz) cultures at 48 DIV. Likewise, we second measured the levels of tubulin acetylation. We found decreased MAP2 levels in the untreated mutant FUS pool compared to untreated Ctrl (Figure 6c), whereas SMI32 levels remained unaltered (Figure 6d). Remarkably, mutant FUS MNs responded to our MS and exhibited elevated MAP2 reverted back to Ctrl levels, whereas Ctrl MNs showed only a minor trend toward increased MAP2 levels, albeit not significant (Figure 6c). MS did not significantly impact SMI32, but it also showed a non-significant trend (Figure 6d and Appendix A). Taken together, the rescue from diminished MAP2 levels in mutant FUS through our MS points to an improvement of microtubule stability that is feasibly beneficial for deficient axonal organelle trafficking and regeneration [13]. For individual lines, refer to Appendix A.

As for post-translationally modified tubulin, our immunofluorescent staining using a specific antibody against acetylated tubulin revealed no change when normalized by total α–tubulin in the untreated mutant FUS versus Ctrl pool, nor after MS (Figure 7).

## 4. Discussion

ALS is a hitherto incurable, fatal motoneuronal disease, urging novel, efficient therapeutic methods. The influence of magnetic stimulations on neuronal diseases and regeneration has been widely investigated for decades, yet their credibility and efficiency remain controversial, especially in the peripheral nervous system. In this study, we intended to investigate the therapeutic potential of MS on peripheral (lower) motor neurons to restore pathological defects of human iPSC-derived spinal MNs from fALS patients with defined mutations in the FUS gene. We revealed a remarkable potential of MS in restoring mutant FUS-mediated phenotypes, e.g., axonal trafficking and axonal regeneration deficits without harming wild-type cells or inducing obvious other detrimental side effects. We observed that magnetic stimulation (MS), including a 10 Hz square-wave excitation, was superior to 2 Hz, thereby suggesting that the higher frequency was more efficient in exerting beneficial effects. The mechanism behind these beneficial effects might arise from microtubule stabilization.

Mutant FUS can cause protein aggregation in the cytoplasm and axoplasm, thereby interfering with organelle trafficking and inducing oxidative stress [43]. Deficient axonal organelle transport in MNs in ALS was described in mutations of C9ORF72, SOD1, TDP-43 and FUS [13,21,44,45]. This fits very well with the “dying back” of MNs in ALS [46]. Thus, we very much focused on this central mechanism in ALS to reveal the putative therapeutic potential of MS in ALS.

First, we wanted to investigate whether our MS protocol alters motoneuronal differentiation per se. We found similar fractions of MAP2- and SMI32-positive neurons in our MS-treated cultures compared to the untreated controls. At first glance, this finding seems to contradict a previous report on murine neural stem/progenitor cells (NSCs) isolated from brain cortices of newborn mice [31]. These authors reported a boosting effect of their MS protocol (1 mT, 50 Hz) on the neuronal differentiation of their NSC model, leading to an increase in TUJ1- and MAP2-positive cells. The discrepancy to our findings can be feasibly explained by (i) the different model systems (primary murine NSCs versus human iPSC-derived spinal MNs), (ii) the different developmental stages (progenitor cells in early neuronal differentiation during the MS treatment versus matured MNs) and (iii) the difference in the technical settings of the MS (1 mT at 50 Hz versus 10 mT at 10 Hz). Notably, we favored a late time point in our MS protocol because this reflected more closely the adult onset of ALS in our disease model than a time point earlier during differentiation. Of note, however, is that iPSC-derived neurons are still of fetal character. Furthermore, our protocols did not yield pure MNs (Figure 1c–e). Therefore, we cannot formerly rule out the presence of intermediate stages from NPCs to mature spinal MNs in our assayed cultures, possibly along with non-neuronal by-products such as S100B-positive flat cells [47]. However, these impurities were restricted to the soma-dendritic proximal MFC compartment, whereas our previous report [13] documented quasi 100% SMI32-positive axons in the distal MFC compartment where all readouts for this study were carried out. Thus, we believe that due to the specific architecture of our compartmentalized culture format, our data still mainly addressed spinal MN biology. In essence, we conclude that our MS protocol most likely did not alter motoneuronal differentiation because our MNs were already sufficiently mature when the MS was performed.

Next, by comparing multiparametric HC profiles of axonal organelle motility, MNs derived from mutant FUS patients and healthy donors were analyzed to assess the rescue of axonal trafficking through exposure to magnetic fields versus untreated ones. Magnetic stimulations had a remarkable capability of restoring mutant FUS-mediated induced phenotypes without harming healthy MNs. Notably, the rescue of organelle motility in the pool of FUS mutant lines depended on the underlying AC frequency. Through exposure to transient magnetic fields, including 10 Hz square waves, we achieved a rescue of impaired mitochondrial and lysosomal organelle motility in distal and proximal axons (Figure 3b,c, Table 2), whereas a lower frequency of 2 Hz only was inefficient (Figure 3a, Table 2).

Distal axonopathy, referred to as retraction of axonal nerve endings at neuromuscular junctions, is a major hallmark of ALS pathology [48]. Impairments of axon branching and growth cone morphology have been described in FUS mutant mouse and drosophila models [39]. Drosophila C9ORF72 ALS models also showed decreased axonal length and growth cone size in spinal motoneurons [49]. Several reported aspects of FUS-ALS pathology relate to the loss of axonal growth cones in FUS mutant motoneurons, including mitochondrial deformation and trafficking, as well as gain-of-toxic function at synapses via FUS aggregation [50,51]. Compromised axonal sprouting leading to enlarged motor units is the cellular mechanism of chronic denervation, a hallmark of ALS. Thus, searching for approaches to stimulate the outgrowth of nerve endings paves the therapeutic path to improved neuroplasticity and regeneration not only in motoneuron diseases but also in others, such as spinal cord injury or peripheral nerve regeneration in general [52]. To this end, we utilized a peripheral axonal injury model through axotomy in microfluidic chambers in order to investigate whether our MS protocol can beneficially impact the advancing new growth cones in MNs derived from healthy donors and FUS-ALS patients (Figure 4a). Since MNs need to first differentiate and mature properly, we had to find a method that enabled us to analyze growth cones after at least 30 DIV. This was not possible in an uncompartmentalized culture format since after 30 DIV and beyond, a highly established, dense soma-dendritic meshwork severely obscures the recognition and tracking of single axonal growth cones. We thus decided to use MFCs, but also, in these, it takes significant time until the axons grow through the microchannels to the distal site, with considerable inter-channel variability. Thus, we chose a late time point (i.e., 48 DIV) at which all channels were mostly penetrated by the axons, regardless of genotype and treatment. At this stage, the distal axons were removed by applying flow shearing forces, leaving the area at the distal channel exit blank. Thereby, we created a defined starting point with easy visualization of the new growth cones, repopulating the blank readout windows here. This experimental setup provided an axonal lesion model to specifically address the ability of late-onset ALS models to regenerate damaged axons, whereas general axonal outgrowth in the course of differentiation and maturation was not captured. This approach appeared attractive in light of an ongoing debate about axonal sprouting and remodeling of the neuromuscular unit in ALS patients (i.e., chronic denervation revealed in electromyography recordings). Furthermore, since neurodegeneration in ALS is supposed to be a “dying back” event (see also [13]), with the first steps of neurodegeneration happening at the distal axon, any kind of improved axonal sprouting might induce functional recovery.

Thus, we established a custom-tailored algorithm to analyze growth cone motility (Figure 4b). The results showed that untreated mutant FUS MNs were significantly slowed down in their axonal regenerative outgrowth compared to Ctrl MNs after axotomy (Figure 4c). The speed of advancing growth cones in mutant FUS MNs through MS at only 10 Hz was significantly ameliorated to the level of Ctrl MNs. Notably, MS did not harm the growth cone dynamics of Ctrl MNs, irrespective of which frequency was used (Figure 4c). Thus, 10 Hz similarly restored axonal regeneration as it does axonal organelle trafficking. Facilitating axonal regeneration has been proposed as a promising therapeutic approach in neurodegeneration in a few studies. Reported candidates were, e.g., microtubule stabilization through Taxol treatment [53] or inhibition of HDAC-6 [54,55] to promote axonal outgrowth. Remarkably, we also found that MS at 10 Hz reverted MAP2 expression back to the wild type level, whereas tubulin expression, posttranslational tubulin modification (i.e., acetylation, Figure 7) or neurofilament expression (SMI32, Figure 6) were unaffected by mutant FUS and MS. In previous studies, MAP2 expression was reported to promote tubulin assembly as well as microtubule stabilization in neurons, which improved the cargo transport of axonal organelles through increased bidirectional motility [33,56]. Interestingly, MAP2 also enhances the bundling of microtubules, thereby supporting motor protein mobility [40,57]. Therefore, the finding of restored MAP2 levels in mutant FUS MNs through MS hints at a possible underlying mechanism of the beneficial effect of MS in our culture system (Figure 6c). Interestingly, approaches for rescuing axonal transport defects through microtubule stabilization were frequently considered putative therapeutic targets in ALS therapy in general but also specifically in FUS-ALS, e.g., by manipulating post-translational modifications in microtubules via silencing HDAC-6 by antisense oligonucleotides in MNs derived from mutant FUS patients [14,55]. Given that the acetylation of tubulin in the FUS mutant lines remained unaltered after MS (Figure 7c), our studies narrowed down the underlying cause of improved axonal organelle motility to the restored MAP2 levels in the mutant FUS pool.

Our axonotomy protocol was inspired by the previously reported method of mechanical axotomy through pipette action [28,29,30]. While a previous report documented no difference in mechanical versus proteolytic (i.e., with Trypsin or Accutase) axotomy in a similar regeneration assay [58], we cannot rule out that the results in a proteolytic axonotomy might be different. Our finding of reduced regenerative outgrowth in untreated mutant FUS-GFP P525L MNs (Figure 4) is consistent with a previous report on cellular murine ALS models [39], though not specifically addressing re-outgrowth after axotomy. The role of ALS-mediating gene mutations in axonal outgrowth and morphology, as well as regeneration after axotomy, was investigated by several other research groups. For example, Akiyama et al. [59] utilized hiPSC-derived spinal MNs bearing an H517D FUS mutation to demonstrate increased axon branching and spine density. Different from our study, neurons were analyzed 10 days after seeding into the final culture format, as opposed to our assays at 48 DIV. Moreover, their analysis was performed closer to the somata with less defined axon specificity due to the lack of compartmentalization. Therefore, because these authors have not performed an axotomy and have not analyzed the speed of growth cones, it remains elusive how their distinct axon branching phenotype relates to our study. Osking et al. [60] utilized cultured murine embryonic and adult primary spinal MNs bearing a G93A SOD1 mutation to demonstrate enhanced neurite outgrowth and branching. Again, this study lacked an axotomy and was performed without compartmentalization, i.e., with unclear axon-specificity, and only 60 µm away from the somata only 2–3 days after final seeding.

Marshall et al. [61] performed an axotomy in a similar cellular model system, i.e., hiPSC-derived spinal MNs cultured in MFCs. Their MNs were bearing an A4V SOD1 mutation and exhibited enhanced axonal re-growth 48 h after axotomy, whereas after just 24 h (our timepoint of analysis) these authors demonstrated no difference from wild type control cells. Moreover, the axotomy was performed 7 days after seeding into MFCs, as opposed to our later time point at 48 DIV. Finally, Garone et al. [58] utilized hiPSC-derived spinal MNs endogenously expressing FUS-GFP P525L, thereby enabling the best comparability to our study, as this particular mutation was contained in our mutant FUS pool. Surprisingly, these authors showed an increase in axonal outgrowth after axotomy in the mutant FUS over its isogenic wild-type control. The reasons for this discrepancy remain elusive. Possible explanations include (i) the different donor patients and, thus, different genetic backgrounds, (ii) different gene editing technologies used (CRISPR-Cas9- in our model versus TALEN-directed mutagenesis in theirs), (iii) different linker sequences between FUS and the GFP-tag and (iv) different developmental stages (maturation of the MNs for just 7 DIV versus our late time point at 48 DIV). The last point is the most obvious difference from our study, since we previously reported that we also do not see axonal trafficking phenotypes in such early differentiation steps, while these progressively develop during longer maturation [13]. In summary, the majority of published reports investigated axonal growth and morphology in much earlier developmental stages, partially even without compartmentalization and in non-human model systems with distinct mutated genes and without axonotomy, which we did not analyze. These studies suggest enhanced axon branching as a common denominator during the early stages of axonal outgrowth not only in familial ALS but also in a zebrafish model of spinal muscular atrophy [62]. In this regard, our study at a much later stage (48 DIV after final seeding into MFCs) appears to be rather different from all other studies and its relation to the abovementioned reports at earlier stages needs to be clarified in future studies.

While we regard the restoration of axonal transport defects through MS in our in vitro ALS disease model as encouraging proof of concept for a feasible novel therapeutic strategy in ALS, there is a clear need for detailed follow-up studies to substantiate our findings. These include—among others—(i) to investigate the optimal technical parameters of the applied magnetic field with respect to AC frequency, orientation, field strength, wave form, etc., (ii) to deepen our understanding of the cellular response of the different magnetic stimuli, including systematic ultrastructural analysis of distal axon before and after MS, (iii) to investigate the underlying mechanisms, e.g., in transcriptome/proteome studies, (iv) to investigate axonal regeneration and the effect of MS on proteolytic (i.e., with Trypsin or Accutase) axotomy, (v) to test our MS protocol on other disease models as well (Parkinson, Huntington’s, Alzheimer Disease, etc.) and (vi) to validate and adapt our MS protocol in clinical pilot studies. Point (i) is of particular importance, as the difference between 2 and 10 Hz in this study indicates that the frequency matters; thus, a broader range of frequencies needs to be systematically investigated to determine the optimum of stimulation. Moreover, we found that our 10 Hz-only MS was capable of ameliorating the apoptotic fraction in one but not all mutant FUS lines (Appendix A). By optimizing the technical parameters in our MS protocol, we hope to achieve better beneficial impacts in this and other disease models as well.

## 5. Conclusions

This work provides proof of concept demonstrating that magnetic field treatments are powerful in restoring several hallmarks of ALS pathogenesis recapitulated in our cultured disease model of human iPSC-derived spinal MNs. These hallmarks comprise deficient axonal organelle trafficking, as well as axonal regeneration. Importantly, MS did not harm healthy donor-derived MNs in any of the applied assays. Our data suggest that stabilized microtubules might be one reason for the observed improvements, but further studies are needed to clarify this in more detail. Additionally, long-term and in vivo studies are required to further substantiate the therapeutic potential of magnetic field treatments, not only in ALS but also in peripheral nerve regeneration in general.

## Figures and Tables

**Figure 1 cells-12-01502-f001:**
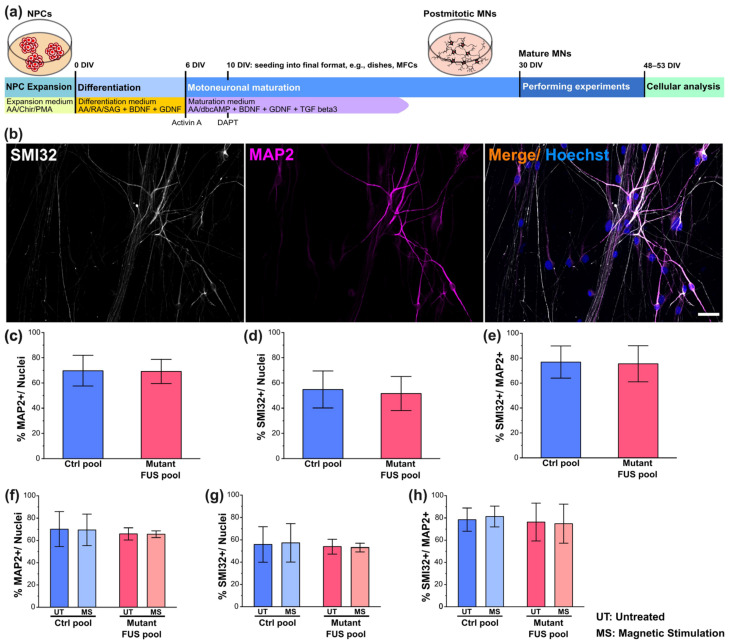
Characterization of iPSC-derived spinal motoneurons (MNs). (**a**) Timeline cartoon of the differentiation pipeline of MNs derived from healthy persons and FUS-ALS patients. AA: Ascorbic acid, Chir: Chiron, PMA: Purmorphamin, RA: Retinoic acid, SAG: Smoothened Agonist (Hedgehog pathway activator, activates Smoothened), dbcAMP: dibutyryl-cyclo-Adenosine monophosphate, BDNF: brain-derived neurotrophic factor, GDNF: glial-cell-derived neurotrophic factor, TGF beta3: tissue growth factor beta3. (**b**) IF microscopy of SMI32 (in white), MAP2 (in magenta) and Hoechst (in blue) of Ctrl MNs. Shown is Ctrl1 (Table 1) as a representative example. Refer to Appendix A for more representative examples from other lines. Scale bar = 40 µm. (**c**–**e**) Quantitative image analysis of panel b of untreated Ctrl (in blue, pooled Ctrl1-3, FUS-GFP WT) and mutant FUS lines (in red, pooled FUS R521L, R521C, P525L, R495QfsX527) (Table 1) revealed no difference in Ctrl versus mutant FUS in any neuronal marker level as expressed as a percentage of marker-positive cells with respect to the entire Hoechst-positive cell population (**c**,**d**) or the entire MAP2-positive neuronal population (**e**). Refer to Appendix A for the individual lines used for the pooled analysis. (**f**–**h**) Same as panel (**c**–**e**) but for magnetically stimulated (MS) versus untreated Ctrl (in blue, pooled Ctrl2 and FUS-GFP WT) and mutant FUS (in red, pooled FUS-GFP P525L and R495QfsX527) (Table 1) MNs. Refer to Appendix A for the individual lines used for the pooled analysis. Note that neither the mutation nor the MS treatment had any impact on any neuronal marker level. Error bars show the SD over all pooled lines and 3–4 individual experiments per line. Ten images per individual line, condition (UT or MS) and experiment were analyzed. One-way ANOVA with Bonferroni post-hoc test revealed no significant alteration in any pairwise comparison.

**Figure 2 cells-12-01502-f002:**
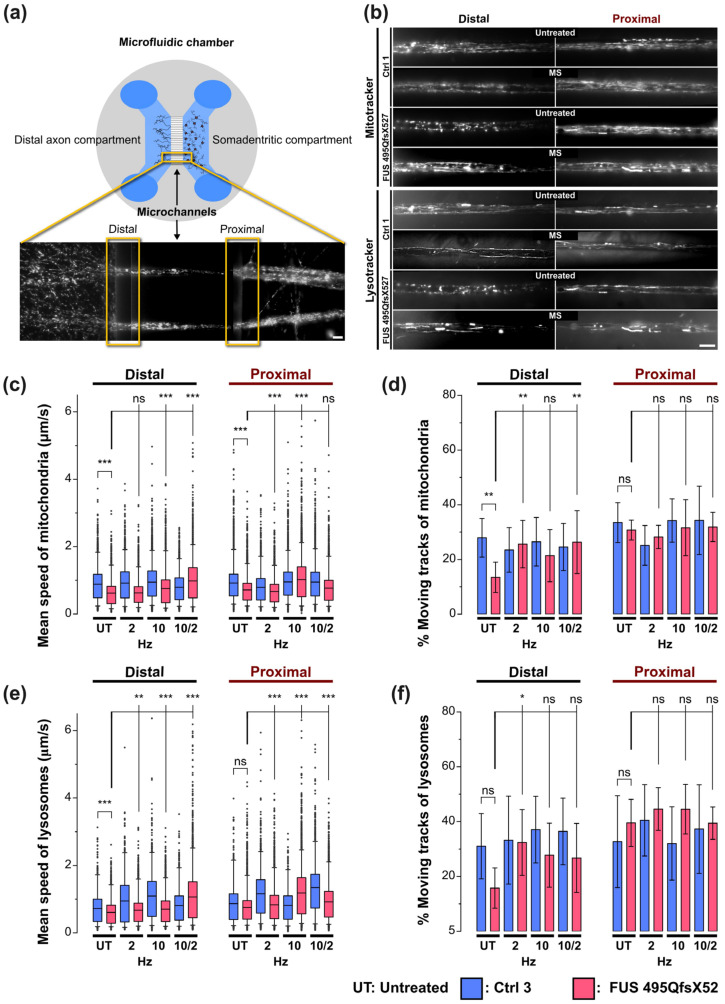
Beneficial impact of magnetic stimulations on axonal trafficking in FUS-ALS. (**a**) MNs derived from healthy persons and FUS-ALS patients were seeded and further differentiated in compartmentalized MFC cultures at 10 DIV to separate axons from the soma-dendritic site. Cartoon illustrating the standardized axonal readout positions distal versus proximal for live imaging at 48 DIV. (**b**) Movies acquired with Mito- and Lysotracker at distal and proximal axonal readout positions are displayed as maximum intensity projections (MIPs) to visualize organelle motility. Corresponds to Appendix A. Note that untreated mutant FUS MNs apparently exhibited decreased mitochondrial and lysosomal motility at the distal site compared to Ctrl MNs, whereas proximal motility remained physiological. Shown are the FUS R495QfsX527 mutant and Ctrl1 as representative examples (Table 1). MS at 10 Hz led to the rescue of these distal trafficking defects in mutant FUS. Scale bar = 10 µm. (**c**) Quantitative tracking analysis of mitochondria as shown in panel (**b**), box plots of track mean speed of individual organelles and (**d**) percentage of moving tracks per movie in the entire movie pool from 3 independent experiments, whiskers span 95% of the entire data sets, center lines represent the means. A one-way ANOVA with the Kruskal–Wallis post-hoc test was utilized to reveal significant differences in pairwise comparisons. Asterisks: highly significant alteration in indicated pairwise comparison, * *p* ≤ 0.05, ** *p* ≤ 0.01, *** *p* ≤ 0.001 and ns: no significant difference. (**e**) Same as panel (**c**) but for lysosomes. (**f**) Same as panel (**d**) but for lysosomes. Refer to Appendix A for more results of the ANOVA. Sample size for panels (**c**,**e**): thousands of organelle tracks. Sample size for panels (**d**,**f**): 45 movies.

**Figure 3 cells-12-01502-f003:**
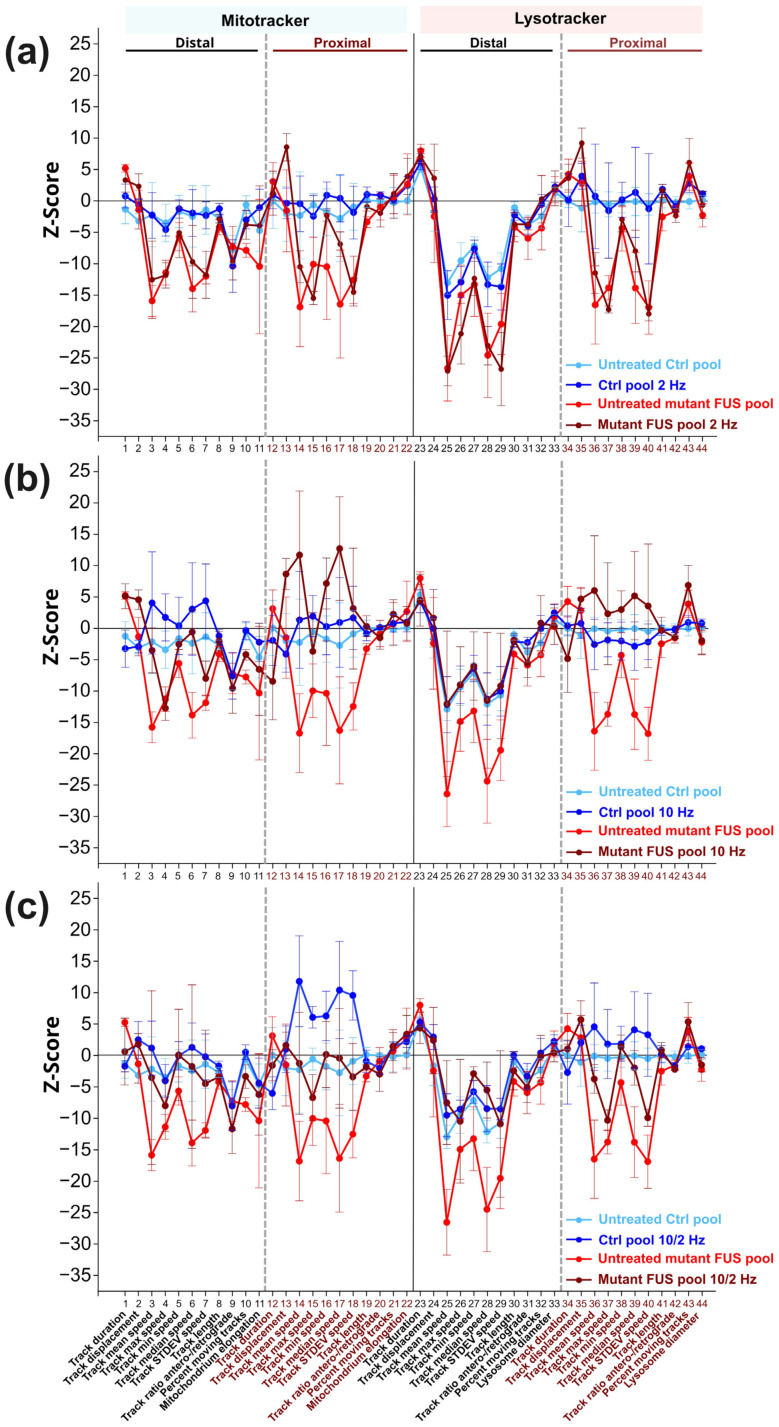
Phenotypic high content (HC) analysis of the beneficial impact of magnetic stimulations (MS) on axonal trafficking in FUS-ALS. (**a**–**c**) Multiparametric HC profiles deduced from the Ctrl (in blue, Ctrl1-3 and FUS-GFP WT, Table 1) and mutant FUS-ALS pool (in red, FUS R521L, R495QfsX527 and FUS-GFP P525L). Refer to Appendix A for the profiles of individual lines. The Z-score deviation of each tracking parameter from the proximal readout of pooled untreated Ctrl lines (in blue) is shown. A set of 11 parameters (bottom labels) was deduced for both the Mito- and Lysotracker, distal versus proximal, as indicated in the header, resulting in 44 parameters in total. (**a**) The impact of MS was assessed at 2 Hz, (**b**) 10 Hz, and (**c**) at their combinatorial sequence of 10 Hz and 2 Hz versus untreated cells (Ctrl pool: compare dark versus light blue profiles, FUS-ALS pool: compare brown versus red profiles). Error bars represent the standard deviation of the respective Z-scores across all pooled lines. Three independent experiments were performed for each line. Sample size for the Ctrl pool: 4 Z-scores from 4 lines per experiment, 12 Z-scores in total. Sample size for the mutant FUS pool: 3 Z-scores from 3 lines per experiment, 9 Z-scores in total. Each Z-score was calculated with thousands of organelle tracks.

**Figure 4 cells-12-01502-f004:**
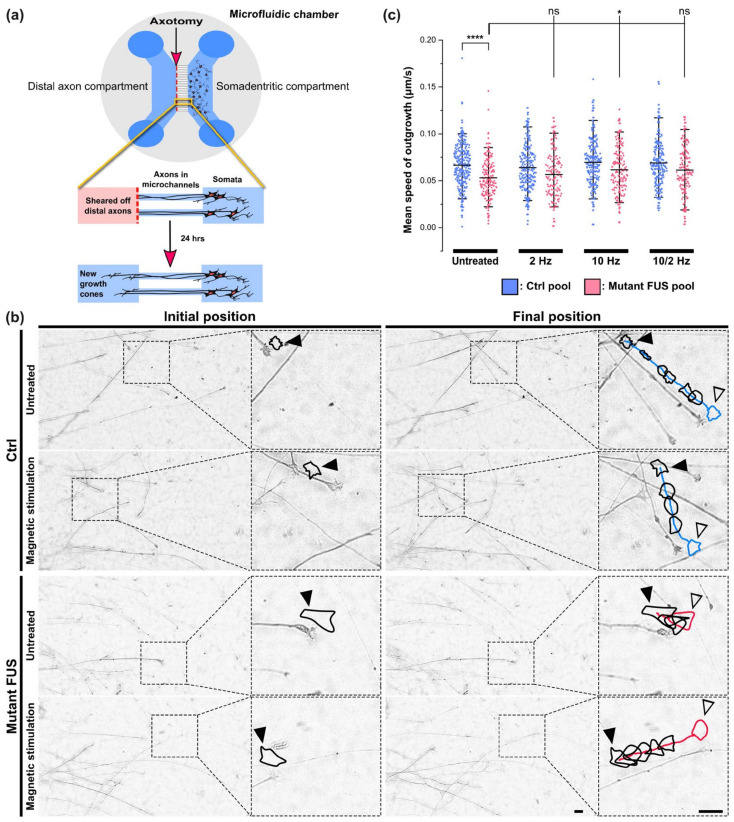
Improvement of axonal regeneration of FUS-ALS in response to magnetic stimulation (MS). (**a**) Cartoon illustrating the in vitro axotomy in MFCs at the distal site followed by new axonal outgrowth of growth cones imaged after 24 h. (**b**) Representative example images (DIC brightfield) of advancing growth cones 24 h after axotomy as shown in panel a on Ctrl 1 versus FUS R495QfsX527 MNs (Table 1) with growth cone contour lines (in black) and colored tracks (blue: Ctrl, red: mutant FUS) highlighted with an offset in order not to obscure the imaged axon. The contoured growth cone of the last movie frame is colored to highlight the final position (blue: Ctrl, red: mutant FUS). Shown are the initial positions of the growth cones (closed black arrowheads) in the first movie frame and their final positions (open black arrowheads) in the corresponding last frame after 2 h of movie recording. Scale bars = 10 µm. (**c**) Quantitative analysis of movies, as shown in panel (**b**). The deduction of mean growth cone speed was performed as described in Section 2.12. Material and Methods. The impact of MS at different frequencies was assessed in four pooled healthy controls and three pooled FUS-ALS patients. Refer to Appendix A for individual lines. N = 3–4 independent experiments. Scatter dot plots display mean speed values of individual growth cones, whiskers show data ranges from 5–95%, and center lines show the populations’ means. A one-way ANOVA with Bonferroni post-hoc test was utilized to reveal significant differences in pairwise comparisons of the different MS treatment conditions (untreated, 2 Hz only, 10 Hz only, 10/2 Hz). Asterisks: highly significant alteration in indicated pairwise comparison, * *p* ≤ 0.05, **** *p* ≤ 0.0001, ns: no significant difference. Sample size (=number of analyzed growth cones) for Ctrl pool untreated: 150, Ctrl pool 2 Hz: 134, Ctrl pool 10 Hz: 137, Ctrl pool 10/2 Hz: 133, mutant FUS pool untreated: 93, FUS pool 2 Hz: 68, FUS pool 10 Hz: 83, FUS pool 2/10 Hz: 78.

**Figure 5 cells-12-01502-f005:**
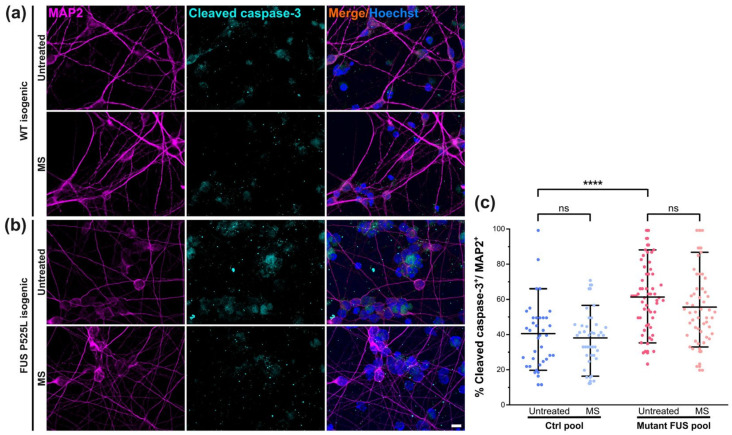
Magnetic stimulation (MS) did not alter the survival of cultured spinal MNs. Immunofluorescence stainings (IF) of MAP2 (in magenta) and the apoptotic marker cleaved caspase-3 (in cyan) in Ctrl (**a**) and mutant FUS (**b**) MNs. Shown as a representative example is the isogenic pair of FUS-GFP WT and FUS-GFP P525L. Scale bar = 10 µm. (**c**) Image quantification of IF stainings as shown in panels a and b of the Ctrl (Ctrl 3 and FUS-GFP WT) and mutant FUS pools (FUS R495QfsX527 and FUS-GFP P525L) to reveal the percentage of cleaved caspase-3-positive cells within the MAP2-positive neuron population, shown as scatter dot plots of per-image values. Refer to Appendix A for the analysis of individual lines. Center lines present the means, whiskers span 95% of all data points pooled from 3 individual experiments of Ctrl3 and FUS-GFP P525L and 2 individual experiments of FUS-GFP WT and FUS 495QfsX527. Ten images per individual line, condition (UT or MS) and experiment were analyzed. Note the significant increase in apoptosis in the untreated mutant FUS pool compared to the untreated Ctrl pool. One-way ANOVA with a post-hoc Bonferroni test was utilized to reveal significant differences in pairwise comparisons. Asterisks: highly significant alteration in indicated pairwise comparison, **** *p* ≤ 0.0001, ns: no significant difference.

**Figure 6 cells-12-01502-f006:**
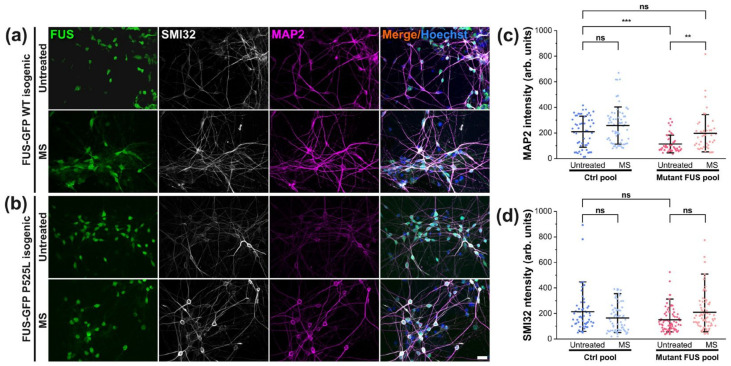
Rescued MAP2 in response to MS in mutant FUS MNs. (**a**,**b**) Immunofluorescent staining of MAP2 (in magenta) and SMI32 (in white) in Ctrl (**a**) and mutant FUS (**b**) MNs. Shown are the isogenic pairs of FUS-GFP WT and FUS-GFP P525L as representative examples. Scale bar = 20 µm. (**c**) Image quantification of panels a and b of the Ctrl (Ctrl1 and FUS-GFP WT, Table 1) and mutant US pool (FUS R495QfsX527 and FUS-GFP P525L); normalized MAP2 intensities per image are shown as scatter dot plots with per-image values. (**d**) Likewise, for SMI32. Refer to Appendix A for individual lines. Note the decreased MAP2 level in untreated mutant FUS, which reverted back to Ctrl levels through MS (panel **c**). Center lines represent the means and whiskers span 95% of the data points from 4 individual experiments. Ten images per individual line, condition (UT or MS) and experiment were analyzed. A one-way ANOVA with a post-hoc Kruskal–Wallis test was utilized to reveal significant differences in pairwise comparisons. Asterisks: highly significant alteration in indicated pairwise comparison, ** *p* ≤ 0.01, *** *p* ≤ 0.001, ns: no significant difference.

**Figure 7 cells-12-01502-f007:**
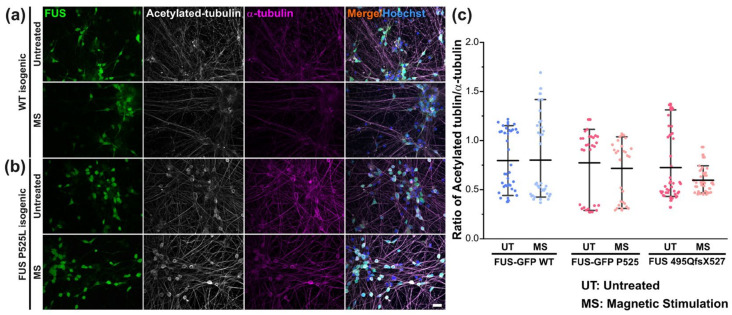
Unaltered tubulin acetylation in response to MS. (**a**,**b**) Immunofluorescent stainings of acetylated tubulin (in white) and α-tubulin (in magenta) in Ctrl (**a**) and mutant FUS (**b**) MNs. Shown are the isogenic pairs of FUS-GFP WT and mutant FUS-GFP P525L as representative examples. Scale bar = 20 µm. (**c**) Image quantification of panels a and b of the Ctrl (Ctrl1 and FUS-GFP WT, Table 1) and mutant FUS pool (FUS R495QfsX527 and FUS-GFP P525L); levels of acetylated tubulin are normalized by α-tubulin and displayed as scatter plots of per-image values. Center lines represent the means and whiskers span 95% of the data points from 3 individual experiments. Ten images per individual line, condition (UT or MS) and experiment were analyzed. One-way ANOVA with Bonferroni post-hoc test revealed no significant alteration in any pairwise comparison.

**Table 1 cells-12-01502-t001:** Patient/proband characteristics. n.d., described as no data.

	Gender	Age of Biopsy	Written as	Mutation	Clinical Features	Previously Characterized in
Control	Male	34	Ctrl1	-	-	[13]
	Female	n.d.	Ctrl2	-	-	[13]
	Female	53	Ctrl3	-	-	[13]
ALS-FUS	Female	58	FUS R521C	R521C^het^	Spinal	[13]
	Isogenic control of R521C	Same as above	FUS WT	WT-EGFP	n/a	[13]
	Isogenic variant of R521C	Same as above	FUS P525L	P525L-EGFP	n/a	[13]
	Female	65	FUS R521L	R521L^het^	Spinal	[13]
	Male	29	FUS 495QfsX527	R495QfsX527^het^	Spinal	[13]

**Table 2 cells-12-01502-t002:** Statistical analysis of multiparametric HC profiles. Two-way ANOVA with Bonferroni post-hoc test for pairwise comparisons of multiparametric HC profiles to reveal significant differences due to mutant FUS and MS at different frequencies. Asterisks: highly significant alteration in indicated pairwise comparisons, ** *p* ≤ 0.01 and **** *p* ≤ 0.0001, ns = not significant.

Bonferroni’s Multiple Comparison Test	Significant Different	Adjusted *p*-Value	Summary
Ctrl untreated vs. 2 Hz	No	˃0.9999	ns
Ctrl untreated vs. 10 Hz	No	˃0.9999	ns
Ctrl untreated vs. 10/2 Hz	Yes	0.0063	**
mutant FUS untreated vs. Ctrl untreated	Yes	˂0.0001	****
mutant FUS untreated vs. 2 Hz	No	0.9708	ns
mutant FUS untreated vs. 10 Hz	Yes	˂0.0001	****
mutant FUS untreated vs. 10/2 Hz	Yes	˂0.0001	****
mutant FUS 2 Hz vs. Ctrl untreated	Yes	˂0.0001	****
mutant FUS 10 Hz vs. Ctrl untreated	No	0.9482	ns
mutant FUS 10/2 Hz vs. Ctrl untreated	No	˃0.9999	ns
mutant FUS 2 Hz vs. Ctrl 2 Hz	Yes	˂0.0001	****
mutant FUS 10 Hz vs. Ctrl 10 Hz	No	˃0.9999	ns
mutant FUS 10/2 Hz vs. Ctrl 10/2Hz	Yes	0.0084	**

## Data Availability

Not applicable.

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
