# Peer review of "Restoring Axonal Organelle Motility and Regeneration in Cultured FUS-ALS Motoneurons through Magnetic Field Stimulation Suggests an Alternative Therapeutic Approach"

_cells, 2023, doi:10.3390/cells12111502_

Round 1

Reviewer 1 Report

Introduction

The authors have divided the introduction into two parts. The first part describes the genetics of ALS and its aetiology, while the second focuses on magnetic field stimulation. I believe the second part is comprehensive and gives the reader a general context on the subject. outlining the already existing achievements and discoveries in the field of ALS (the ones that the authors themselves report in the results):

-       Axonal organelle transport in ALS

-       Axonal regeneration (outgrowth) 

-       Mitochondria morphology in ALS

-       Cytoskeleton deficit in ALS

Materials and Methods

The authors wrote in detail about the methodology used for their experiments. I don't have any suggestions about it.

Results

3.1 Neuronal characterization and the effect of magnetic stimulations on neuronal differentiation.

The authors characterized iPSC-derived MNs to evaluate the differentiation process by immunostaining analysis in FUS mutant MNs and after magnetic stimulation treatment.

The authors show no differences between control and FUS mutant lines considering the “mean percentage” of MAP2, TUJ1 and SMI32.

Major points

1)    Does the “mean percentage” referred to by the authors relate to signal intensity or spread area? What time point was used to do the experiments? DIV 30?

-       If it is the mean of the signal intensity, there would be an inconsistency with Figure 6, in which the authors show that the MAP2 signal is lower in the mutant FUS MNs.

-       If it is an analysis based on the area that the specific marker covers in each image, I would like to ask the authors to bring the same analysis back to MFC. The authors use a differentiation that does not generate a pure population of MNs. No methodology is mentioned to isolate only motor neuron cells in dishes (such as FACS sorting). When immunostaining against specific cellular tags in a mixed population, nonspecific events may bias the assay. This is evident in figure 6.

2)    This analysis is insufficient to state that there are no differences in the differentiation and maturation of the motor neuron with and without magnetic stimulation treatment. Were studies conducted on the differentiation times of the control and mutant lines? What investigations have been undertaken to affirm no differences in cell maturity?

Minor points

1)    Insert the statistical test used in the caption.

2)    Show in the supplementary figures representative images of the control and mutant about the markers used in the analysis.

3.2 MS Restores deficient axonal organelle transport in FUS-ALS

In this section, the authors investigate any beneficial impact of MS on the motility of organelles in the axon. They perform the test on MNs replated in MFC to detail the study on distal and proximal axons.

It is an interesting experiment that I believe requires some clarification.

Major points

1)    When seeding MNs in microfluidic devices, controlling how many axons enter each channel is impossible. With this assumption, I would ask the authors the following questions:

-       Were the analyses of Mito and Lisytracker conducted by visual field or by single axon? If the former, how many images were captured? Is the error derived from the different number of axons present in the microgrooves negligible? If the analysis was conducted on a single axon, how would the authors distinguish each axon from others in the same microgroove?

Minor points

1)    Insert the statistical test used in the caption.

2)    2) Figures 2c, d, e, f, g, h, and i are too small; it is impossible to read anything, making it even more challenging to understand the data.

3.3 Rescue of axonal regeneration defects in FUS-ALS through MS. 

The authors test the axonal outgrowth of control and mutant motor neurons, suggesting that magnetic stimulation induces a rescue of the axonal defects observed in the mutant line.

Major points

1)    The axotomy performed in the axonal chamber is mechanical. The authors claim to have committed an axotomy using the pipette in the materials and methods paragraphs. A mechanical axotomy is less accurate than a chemical one (e.g. trypsin). This often makes the axotomy incomplete, with possible truncated cuts on the axons but not all at the same height of the microfluidics (to be clear, at the level of the microgrooves) that the authors showed in the cartoon. It would be worth performing chemical damage. 

2)    Are the images shown in figure 3b representative? If it is true, there is an inconsistency between the data displayed and the assumptions made in the first paragraph. The mutation does not affect the cells' ability to differentiate.

However, if the image is not representative, I would like to ask the authors to change the images and show others that describe that the number of axons is comparable.

3)    Not able to control the number of axons entering the microgrooves, how can the outgrowth be discriminated before and after axotomy (with and without treatment) of the same single axon?

3.4 Recovery of compromised mitochondria morphology in FUS-ALS after MS.  

In this section, the authors study the mitochondrial morphology in FUS-ALS MNs before and after MS.

Major points

1)    How many times was the experiment carried out, and on how many lines?

2)    Would it be possible to quantify this evidence? How many mitochondria show an aberrant phenotype from the total number of mitochondria analysed?

3.5 MS did not alter neuronal survival 

The authors test by immunostaining against caspase-3 the levels of apoptosis after treatment with MS.

Major points

1)    Figure 5a shows a precise positioning error in the images related to Cleaved caspase-3 in the WT-isogenic line. Caspase-3 associated with MS is the representative image of untreated. 

2)    How do the authors suggest that MS does not affect the motor neuron population? The authors use a mixed population, so how do they know that caspase-3 refers to MNs? In analysing these data, some fields were acquired with a % of MNs population higher than others. If so, the data would not be representative. By adjusting the focus? Also in this case, there would be no certainty that the caspase is related to the MNs. Indeed, the authors show there are DAPI-positive cells that are not MNs, with a diameter that is sometimes even larger.

3.6 Magnetic stimulations modulated cytoskeleton integrity in MNs with FUS mutations. 

Major points

1)    FUS is an RNA-binding protein that localises in the nucleus in the WT condition. In contrast, there is an evident mislocalisation from the nucleus to the cytoplasm in the NLS mutant condition. Assuming that not all iPSC-derived MNs models show the same percentage of mislocalised FUS in the cytoplasm, how can the authors observe nuclear FUS in the P525L FUS mutant and diffuse FUS in WT? Is there an error associating names with images? Or is it a signal oversaturation problem?

2)    Assuming the image of Figure 6 was representative. The authors state a lower expression of MAP2 in the FUS mutant and that MS treatment restores the physiological condition. Why is it impossible to appreciate the same phenotype in the image figure 5b?

3)    Assuming again that the image of Figure 6 is representative. I need some clarification on what is asserted in the first paragraph.

Discussion

Some articles already published describe different results from those shown by the authors. Some of them should be included and argued. They should explain why they see different phenotypes. Different differentiations? Different analysis times? What is the maturity level of other motor neurons?

General comments. 

Referring to the mutant FUS pool instead of the FUS pool would be more appropriate. All lines used in this article express the FUS protein. The difference lies in the expression of a WT or mutated protein.

Being human cells, the correct nomenclature provides for writing proteins with capital letters (MAP2 instead of Map2).

Author Response

Dear Editor, dear reviewers,

we are very grateful for the many constructive suggestions raised by the three reviewers. As you can find below, we tried hard to adress and answer every single comment. We believe that this significantly strengthens our manuscript and hope that it is now acceptable for publication.

Comments and Suggestions for Authors

Introduction

The authors have divided the introduction into two parts. The first part describes the genetics of ALS and its aetiology, while the second focuses on magnetic field stimulation. I believe the second part is comprehensive and gives the reader a general context on the subject, outlining the already existing achievements and discoveries in the field of ALS (the ones that the authors themselves report in the results):

-Axonal organelle transport in ALS

-Axonal regeneration (outgrowth) 

-Mitochondria morphology in ALS

-Cytoskeleton deficit in ALS

Response: we deeply thank the reviewer for this positive feedback.

Materials and Methods

The authors wrote in detail about the methodology used for their experiments. I don't have any suggestions about it.

Response: We thank the reviewer for this positive feedback, we did however add some informations especially specifying e.g. position of and more details on live cell imaging, readouts, IF image quantificaton etc. (see comments below of other reviewers).

Results

3.1 Neuronal characterization and the effect of magnetic stimulations on neuronal differentiation.

The authors characterized iPSC-derived MNs to evaluate the differentiation process by immunostaining analysis in FUS mutant MNs and after magnetic stimulation treatment.

The authors show no differences between control and FUS mutant lines considering the “mean percentage” of MAP2, TUJ1 and SMI32.

Major points

1) Does the “mean percentage” referred to by the authors relate to signal intensity or spread area? What time point was used to do the experiments? DIV 30?

Response: „Mean percentage“ means, e.g., the number of MAP2-positive cells per total cell count in the image (by HOECHST33342). There was no relation to signal intensity or spread area. All analysis were performed on DIV 48, as written in the M&M part 2.5 and in the Fig1a.

-If it is the mean of the signal intensity, there would be an inconsistency with Figure 6, in which the authors show that the MAP2 signal is lower in the mutant FUS MNs.

Response: We did not measure the signal intensity in Figure 1, see above.

-If it is an analysis based on the area that the specific marker covers in each image, I would like to ask the authors to bring the same analysis back to MFC. The authors use a differentiation that does not generate a pure population of MNs. No methodology is mentioned to isolate only motor neuron cells in dishes (such as FACS sorting). When immunostaining against specific cellular tags in a mixed population, nonspecific events may bias the assay. This is evident in figure 6.

Response: As we said above, the analysis in Fig. 1c-j is not based on the area that a specific marker covers in each image, but instead on the number of positive cells per total cells. The decision whether a cell is positive or not was determined by using an unbiased standard thresholding tool of the FIJI software. We added further details to M&M about our IF image analysis in section 2.7. for more clarity.

2) This analysis is insufficient to state that there are no differences in the differentiation and maturation of the motor neuron with and without magnetic stimulation treatment. Were studies conducted on the differentiation times of the control and mutant lines? What investigations have been undertaken to affirm no differences in cell maturity?

Response: We agree with the reviewer, but want to point out that we used a standard differentiation protocol throughout the last years and that we showed in our previous publications that our MNs (WT and Mut) are already fully matured at the timepoints investigated in the current study. Using this protocol, we have already shown in e.g. Naumann et al. 2018 (PMID: 29362359 ) that our MNs are fully matured by this time and show no difference in (moto)neuronal marker stainings including Hb9, Islet and ChAT at 30-60 DIV in Mut vs WT, and that these neurons are also electrophysiologically depicting typical hallmarks of mature neurons (Naumann et al., 2018 - PMID: 29362359, Naujock et al., 2016 - PMID: 26946488, Bursch et al, 2019 - PMID: 31108504). We also showed that neurons develop normally but then aquire functional deficits (starting at DIV21) followed by structural axonal degeneration (DIV60) and cell death (DIV 110), see Naumann et al for details (PMID: 29362359). Details on microfluidics and cell differentiation of those were also published several times by us, (e.g. PMID: 32151030, PMID: 29362359, PMID: 32084385, PMID: 30422121). We therefore believe that our end point analysis here at 48 DIV is sufficient to reveal whether the MS from 30-48 DIV alter the motoneuronal fraction or not and wanted also to avoid not to repeatedly present known facts on the one hand side. On the other side, our study is a first proof of concept study of this kind of magnetic stimulation of patient-derived motoneurons including ALS patients, thus it is more concept- and hypothesis-generating. Therefore, follow-up studies are clearly warranted to (i) validate the results in more ALS cell lines bearing, e.g., mutant SOD1, C9orf72, TDP43 and other disease models (HD, AD, PD, etc.), (ii) to further „titrate“ the most efficient frequencies/stimulation conditions, (iii) to increase our understanding of the cellular responses to the different magnetic simuli and (iv) also to further understand the underlying mechanisms. Thus, we toned down our statement and did also include a “limitaton of the study“ and “future outlook” paragraph in the end of the discussion (see also reviewer #2&3).

Minor points

1) Insert the statistical test used in the caption.

Response: We did so.

2) Show in the supplementary figures representative images of the control and mutant about the markers used in the analysis.

Response: We added to the supplemental part a new figure S1 to provide more representative examples from cell lines used for the pooled analysis in the corresponding figure 1b-f. Likewise, we added a new figure S3 to provide more representative examples from cell lines used for the pooled analysis in the corresponding figure 1g-j.

3.2 MS Restores deficient axonal organelle transport in FUS-ALS

In this section, the authors investigate any beneficial impact of MS on the motility of organelles in the axon. They perform the test on MNs replated in MFC to detail the study on distal and proximal axons.

It is an interesting experiment that I believe requires some clarification.

Major points

1) When seeding MNs in microfluidic devices, controlling how many axons enter each channel is impossible. With this assumption, I would ask the authors the following questions:

-Were the analyses of Mito and Lisytracker conducted by visual field or by single axon? If the former, how many images were captured? Is the error derived from the different number of axons present in the microgrooves negligible? If the analysis was conducted on a single axon, how would the authors distinguish each axon from others in the same microgroove?

Response: We appreciate this comment. A single axon-based algorithm is simply not possible in MFCs due to limitation of the optical resolution within the densely packed axon bundle in the microchannels. Assignments to single axons would require ultra low density cultures, thereby hampering a good statistic bulk yield due to the low number of organelles captured. And these low densities might also influence the general biology of MNs (as MFCs might as well). Thus, we decided to use MFCs at higher cell density here. In that, it is true that the number of axons growing through a microchannel can and do variate. Since we used, however, strictly standardized readout positions distal vs proximal in the channels for imaging and standardized semi-automated quantification paradigms, inter-channel variability is averaged out due to the pooling of thousands of tracks across many channels and movies. (see ,e.g., PMID: 32151030, PMID: 29362359, PMID: 32084385, PMID: 30422121). We added additional informations about these points in the revised M&M section:

  1. At least 5 movies per readout position and MFC, resulting in at least 15 movies per cell line and condition and experiment due to three technical replicas (=MFCs), at least three independent experiments were performed on independent differentiation pipelines.
  2. The analysis is done on a per-organelle-base and does not require any assigment to a particular axon within the axon bundle in the channel to obtain the bulk statistics shown in figure 2 and 3, organelle tracks are pooled across all movies of a condition.
  3. Different axon numbers across the channels does not introduce an error per se, inter-channel variability is averaged out due to the pooling of thousands of tracks across many channels and movies. In essence, we never observed that the organelle motility in a bundle composed of, e.g., 10 axons is different from a bundle composed of 20 axons. Any inter-channel variability here is just part of he overall variability in the whole pooled organelle population.

Minor points

1) Insert the statistical test used in the caption.

Response: We did so.

2) Figures 2c, d, e, f, g, h, and i are too small; it is impossible to read anything, making it even more challenging to understand the data.

Response: We rearranged figure 2 completely by splitting it into two new figures, thereby increasing the viewing size of all box plots (panel 2c and e), bar graphs (panel 2d and f) and HC profiles (panel 3a-c). In the new figure 2, we carefully revised the significance indicators on top (showing asterisks and “ns”) to provide a better logic and consistent pattern. Moreover, we added a new Table S1 to the supplemental part providing more details about the ANOVA. In the new figure 3, we carefully revised the line colors and thickness to better discern the untreated from the magnetically treated conditions. The Ctrl pool is always shown in light blue and its treated counterpart in dark blue. Likewise, the mutant FUS pool is always shown in red and its treated counterpart in brown. In essence, the simplified logic here is to compare the darker shade of a given color against its lighter shade to figure out the effect of the magnetic treatment. We hope this improved layout conveys the take-home message of the multiparametric HC analysis at better clarity.

3.3 Rescue of axonal regeneration defects in FUS-ALS through MS. 

The authors test the axonal outgrowth of control and mutant motor neurons, suggesting that magnetic stimulation induces a rescue of the axonal defects observed in the mutant line.

Major points

1) The axotomy performed in the axonal chamber is mechanical. The authors claim to have committed an axotomy using the pipette in the materials and methods paragraphs. A mechanical axotomy is less accurate than a chemical one (e.g. trypsin). This often makes the axotomy incomplete, with possible truncated cuts on the axons but not all at the same height of the microfluidics (to be clear, at the level of the microgrooves) that the authors showed in the cartoon. It would be worth performing chemical damage. 

Response: On a side note: due to the insertion of the new figure 3 (HC profiles), the axotomy data are now in figure 4. The aims of the experiment were to compare the ability of axonal regeneration in healthy versus FUS-ALS patients‘ neurons and whether the effect of magnetic stimulations can rescue the defect of regeneration after the injury. The reviewer is correct that we used a mechanical way to perform the axotomy. Even in the case it would yield an incomplete axotomy, this does not affect our quantifications since we quantified the regrowth in a defined time from a defined starting position to a defined end position.

In contrast, we believe that chemical axotomy is not feasable in the MFC setup: We doubt that an enzymatic axotomy would result in a sharp truncation boundary at the distal channel exit at all because, e.g., trypsin will penetrate the microchannels during its incubation time to various extents depending on the local microflow, axon bundle density, obstacles etc. Therefore. each channel could potentially have a different truncation boundary. Moreover, the trypsin treatment would probably digest the crucial laminin coating in our MFCs, thereby hampering any new outgrowth. For these reasons, we disfavour proteolytic approaches. Instead, our mechanical shearing flow protocol always leaves the channel exit area (where the subsequent imaging is then performed) blank with the remaining truncated axon trunks inside channels reaching very close to their distal end as we verified by careful viewing right after the axotomy. We revised the discussion to make this point accordingly (line 952).

2) Are the images shown in figure 3b representative? If it is true, there is an inconsistency between the data displayed and the assumptions made in the first paragraph. The mutation does not affect the cells' ability to differentiate.

However, if the image is not representative, I would like to ask the authors to change the images and show others that describe that the number of axons is comparable.

Response: We agree and exchanged the images to have better representative examples in the revised figure 4b. We provide now overview images for each condition depicting better similar axon density in the newly populated distal areas after axotomy whereas the inlets show magnifications to illustrate the difference in axon outgrowth of single growth cones and its rescue through MS.

3) Not able to control the number of axons entering the microgrooves, how can the outgrowth be discriminated before and after axotomy (with and without treatment) of the same single axon?

Response: There might have been a misunderstanding. We did not intend to investigate whether there are differences in outgrowth behaviour with and without axotomy. There were two reasons why we used this setup that enabled us two address two questions. First, we wanted to analyse axonal growth in WT vs. mutants and whether this can be beneficially affected by MS. Since MNs need first to differentiate and mature properly (see discussion above), we had to find a way of being able to analyse growth cones after at least 30 DIV. This is not possible in an uncompartmentalized culture format, since after >30 DIV one has a highly established neurite network in which single axonal growth cones cannot be identified in the dense soma-dendritic meshwork. We thus decided to use MFCs. But also in these it takes significant time until the axons grow through the microchannels, and there are significant inter-channel variations. Thus, we needed to have a setup in which we can let the MNs mature properly and then create a defined starting point of axonal growth cones. This let us decide for the mechanical axonotomy model at the distal exit side of the MFCs, which is known to only show axonal outgrowth and, using 900µm channels, these were purely MNs (PMID: 32151030, PMID: 29362359).

The second reason was that we intended to use this axonotomy model to investigate whether FUS-ALS mutant MNs are impaired in the axonal re-growth and whether this might be influenced beneficially by MS. This is an important question also in ALS patients, since there is a debate about axonal sprouting and remodelling of the neuromuscular unit (i.e. chronic denervation revealed in electropmyography recordings). Furthermore, since neurodegeneration in ALS is supposed to be a “dying back” event (see also PMID: 29362359, here Fig. 2 for FUS-ALS), with the first steps of neurodegeneration happening at the distal axon, any kind of improved axonal sprouting might induce functional recovery. These are the reasons why we utilized this setup here.

3.4 Recovery of compromised mitochondria morphology in FUS-ALS after MS.  

In this section, the authors study the mitochondrial morphology in FUS-ALS MNs before and after MS.

Major points

1) How many times was the experiment carried out, and on how many lines?

2) Would it be possible to quantify this evidence? How many mitochondria show an aberrant phenotype from the total number of mitochondria analysed?

Response: We totally agree with the reviewer and have to say that we intensively discussed this part prior submission and once more during revisions. The experiments were technically demanding and there had been many experiments in which cells were lost or significantly altered during the prosessing for the EM. This is the reason why we only obtained reliable results from 1 control and 1 mutant cell line. We were and are still also concerned about quantifying the images, since there are some limitation to investigate the peripheral mitochondria inside axons using the presented technology, which are extremely fragile and one never can guarantee how many are lost during sample processing. Therefore, it was difficult to get enough mitochondria to quantify and thus we wanted to avoid any misjudgements. Along with the reviewer’s sorrow we decided to relocate these results to the supplements (now figure S7) clearly stating that these are very preliminary to avoid any kind of overemphasizing.

3.5 MS did not alter neuronal survival 

The authors test by immunostaining against caspase-3 the levels of apoptosis after treatment with MS.

Major points

1) Figure 5a shows a precise positioning error in the images related to Cleaved caspase-3 in the WT-isogenic line. Caspase-3 associated with MS is the representative image of untreated. 

Response: We thank the reviewer for having keenly spotted this mistake, we corrected this. Please accept our appologies.

2) How do the authors suggest that MS does not affect the motor neuron population? The authors use a mixed population, so how do they know that caspase-3 refers to MNs? In analysing these data, some fields were acquired with a % of MNs population higher than others. If so, the data would not be representative. By adjusting the focus? Also in this case, there would be no certainty that the caspase is related to the MNs. Indeed, the authors show there are DAPI-positive cells that are not MNs, with a diameter that is sometimes even larger.

Response: The analysis was restricted to MAP2-positive neurons, but not to motoneuron markers. Thus we can clearly state that there is no increased neuronal cell death but cannot specifically state about the motoneuronal cell death. But at least we are confident to have captured the motoneuronal population together with other possible neuronal subtypes which we cannot formerly rule out. We revised M&M, section 2.7., and Results, section 3.5. (line 867) accordingly.

3.6 Magnetic stimulations modulated cytoskeleton integrity in MNs with FUS mutations. 

Major points

1) FUS is an RNA-binding protein that localises in the nucleus in the WT condition. In contrast, there is an evident mislocalisation from the nucleus to the cytoplasm in the NLS mutant condition. Assuming that not all iPSC-derived MNs models show the same percentage of mislocalised FUS in the cytoplasm, how can the authors observe nuclear FUS in the P525L FUS mutant and diffuse FUS in WT? Is there an error associating names with images? Or is it a signal oversaturation problem?

Response: It is true that FUS is normaly mainly localized in the nucleus and in case of FUS-NLS mutations there is an increase in the amount of cytoplasmic FUS and reduction of nuclear FUS, however, only rarely a complete loss of nuclear FUS. The latter might be more obvious in post mortem studies, but in iPSC-derivates there is typically some remaining nuclear FUS even in very strong NLS mutants (such as the P525L mutant). Moreover, we have to admit that in the GFP-tagged lines one can find also increased GFP signals in the cytoplasm in case of wildtype FUS, but which is still below the level of the mutants. Thus, we agree with the reviewer that the choice of our images together with a signal oversaturation to depict also the cytoplasmic FUS led to a not perfectly representative image here. We did correct this. Finally, we wish to point out that the optical magnification of these images here (x20) is not well suitable for assessing the nucleo-cytosolic partitioning of FUS and that this was not the biological question we wished to address in Fig. 5. Rather, we wished to provide a sufficient overview of the cytoskeletal markers and the neuritic network here. From this perspective, the FUS channel is a bit redundant here.

2) Assuming the image of Figure 6 was representative. The authors state a lower expression of MAP2 in the FUS mutant and that MS treatment restores the physiological condition. Why is it impossible to appreciate the same phenotype in the image figure 5b?

Response: Our apologies for having not perfectly representative images so far. We replaced them in the revised version of the manuscript.

3) Assuming again that the image of Figure 6 is representative. I need some clarification on what is asserted in the first paragraph.

Response: We did so, see above.

Discussion

Some articles already published describe different results from those shown by the authors. Some of them should be included and argued. They should explain why they see different phenotypes. Different differentiations? Different analysis times? What is the maturity level of other motor neurons?

Response: We would like to popint out that we already tried to discuss recent literature on MS, as we did e.g. in line 527 (section 3.1., Results, reference 31) in the original version. In the revised manuscript, we illuminated more on this discrepant report (Discussion, line 980 and following). For example, the respective reference was using murine cells early in neuronal differentiation whereas we were using iPSC-derived human neurons at late endpoints when cells were already fully matured, etc.

General comments. 

Referring to the mutant FUS pool instead of the FUS pool would be more appropriate. All lines used in this article express the FUS protein. The difference lies in the expression of a WT or mutated protein.

Being human cells, the correct nomenclature provides for writing proteins with capital letters (MAP2 instead of Map2).

Response: We thank the reviewer for having keenly spotted these mistakes and corrected them throughout the manuscript accordingly.

Reviewer 2 Report

This manuscript has many important information missing that prevented me from evaluting it. The following comments highlights what is missing. I will be happy to reevaluate the manuscript after providing the missing information. 

1) Blinding and bias - many subjective decisions are made by the nature of the analysis. Even the automated algorithms are set up with user-selected ROI, user-selected threshold and other image filtering parameters, user-chosen cone to follow in some cases, etc. All of these elements could be subject to human error and bias without blinding and also subject to drift without analysis order randomization and image acquisition randomization. Please provide details on randomization and blinding that was practiced. Also, please clarify if all images were collected and analyzed by the same experimenter?

2) Figure 1: Marker level is the percentage of positive cells, but how is a cell identified as positive or negative? Is this a manual, subjective decision? And if so, was the decision maker blinded to the experimental group?

3) Figure 2: This figure is extremely difficult to interpret. Panels c and d: how were certain comparisons chosen to be illustrated with a significance line and others not? Are all pairs with no bars assumed to be NS? Panels d and f: some pairs are labeled as NS while other pairs have no significance line. Are we to assume the other comparisons did not show significance as well? Panels g-h: it is extremely difficult to differentiate the dotted and solid lines. Major revisions needed to this figure. At minimum, blow up a region of the figure showing the major take away message.

4) Table 2: I would like to see a row in this table comparing FUS 10/2 to Ctrl 10/2. Also, why was one group chosen to have the last session be 2 Hz while all the other sessions were 10 Hz? Was this just a random guess, an experimental mistake, or an informed decision?

5) Rescue of axonal regeneration defects in FUS-ALS through MS: How does regrowth after cutting the axons relate to disease changes in ALS? The axons are not cut in the disease…

6) Please speculate why 10/2 Hz shows the best organelle motility recovery (but not 10 Hz or 2 Hz alone), but 10 Hz only impacted cone regrowth. 

7) Figure 4: Were more images analyzed or only those shown in the figure? Did all images show this same phenomenon? I would like to see at least a basic quantification here, since this figure seems to be the basis for the “therapeutic potential” which is highly emphasized in the title, abstract, and conclusion of the paper. At minimum, how many images clearly showed the illustrated changes vs how many showed no changes vs how many showed alternative changes.

8) Please provide a citation or other justification for using fold-change in figures 5-7. How are the raw intensity values measured for each image? 

9) Statistics: Supplemental data with detailed results of all statistical comparisons should be provided.

10) “Ditto” is a very informal word and should be replaced with “likewise” or something similar.

Author Response

Dear Editor, dear reviewers,

we are very grateful for the many constructive suggestions raised by the three reviewers. As you can find below, we tried hard to adress and answer every single comment. We believe that this significantly strengthens our manuscript and hope that it is now acceptable for publication.

Comments and Suggestions for Authors

This manuscript has many important information missing that prevented me from evaluting it. The following comments highlights what is missing. I will be happy to reevaluate the manuscript after providing the missing information. 

1) Blinding and bias - many subjective decisions are made by the nature of the analysis. Even the automated algorithms are set up with user-selected ROI, user-selected threshold and other image filtering parameters, user-chosen cone to follow in some cases, etc. All of these elements could be subject to human error and bias without blinding and also subject to drift without analysis order randomization and image acquisition randomization. Please provide details on randomization and blinding that was practiced. Also, please clarify if all images were collected and analyzed by the same experimenter?

Response: All images were collected and analyzed by the same experimenter. However, most if not all significant differences became visible not until the analysis was performed and most of the setups anyway could not be judged without a systematic quantifications. Blinding was anyway difficult since the FUS-GFP clearly is different in FUS mutant vs. FUS wt, therbey deblinding it automatically. To minimize any bias, all experiments were analysed using strictly standardized, pre-defined readout positions and settings followed by semi-automatic image analysis.

We included the following statement in the revised manuscript (M&M, section 2.14.): “All experiments were analyzed by the same experimenter using strictly standardized, pre-defined readout positions and settings followed by semi-automatic image analysis. Even not blinded (since the intracellular FUS-GFP pattern is clearly different in mutant FUS versus WT FUS, thereby deblinding the genotype automatically), these standardized procedures ensured the minimization of any bias in the quantification procedure.“

2) Figure 1: Marker level is the percentage of positive cells, but how is a cell identified as positive or negative? Is this a manual, subjective decision? And if so, was the decision maker blinded to the experimental group?

Response: Blinding was difficult since the FUS-GFP is clearly different in FUS mutant vs. FUS wt, deblinding it automatically. Indeed the analysis in figure 1 was manual and thus in the end a bit subjective, but since the manual counting was performed after thresholding in FIJI (see revised material and methods, section 2.7.) we regard the possible bias as neglectable. Moreover, at least we blinded for the condition (treatment vs. non treatment). We amended M&M, section 2.7., to explain our IF image quantifications in more detail.

3) Figure 2: This figure is extremely difficult to interpret. Panels c and d: how were certain comparisons chosen to be illustrated with a significance line and others not? Are all pairs with no bars assumed to be NS? Panels d and f: some pairs are labeled as NS while other pairs have no significance line. Are we to assume the other comparisons did not show significance as well? Panels g-h: it is extremely difficult to differentiate the dotted and solid lines. Major revisions needed to this figure. At minimum, blow up a region of the figure showing the major take away message.

Response: As also written for Reviewer#1 (see above for more detailed response about this point), we agree with the reviewers and did rearrange figure 2. We actually split it in two figure (Fig 2&3 of the revised manuscript) and also thoroughly revised its layout for better clarity. We also added details about statistics in the revised supplement in a new Table S1.

4) Table 2: I would like to see a row in this table comparing FUS 10/2 to Ctrl 10/2. Also, why was one group chosen to have the last session be 2 Hz while all the other sessions were 10 Hz? Was this just a random guess, an experimental mistake, or an informed decision?

Response: We appreciate the comment. We added the requested new row and some more to Table 2. The presented paradigms were neither an experimental mistake nor a random guess. In our search for a meaningful frequency setting for our MS experiments, we encountered a growing number of published reports about benefical effects of extremely low frequency electromagnetic field (ELF-EMF) applications in the range of 1-100 Hz. In essence, it was difficult for us to decide which of the already tested frequencies could serve as the most promising and meaningful starting point for our own MS experiments on human iPSC-derived MNs. This was simply due to the multitude of different disease models and biological readout assays tested with unclear relevance for our own model system and biological question we wished to address. Thus, in order to streamline our proof of concept study and to avoid tedious screenings to empirically establish efficient frequency settings, we decided to use AC rectangular instead of sine waves. Unlike the homogenous sine waves, rectangular waves have the benefit of producing a spectrum of higher harmonic frequencies according to the Fourier analysis (see also Material and Methods), thereby allowing to test multiple frequencies simultaneously. For example, a basal frequency of 2 Hz rectangular waves results in 2 Hz of amplitude 1 + 6 Hz of amplitude 1/3 + 10 Hz of amplitude 1/5 + 14 Hz of amplitude 1/7 + etc. Likewise, a basal frequency of 10 Hz results in 10 Hz of amplitude 1 + 30 Hz of amplitude 1/3 + 50 Hz of amplitude 1/5 + 70 Hz of amplitude 1/7 + etc. We decided to test 2 Hz to capture the low ELF-EMF range and 10 Hz for the higher range. To counterbalance the decreasing amplitude at higher harmonic frequencies, we utilized a relatively high field strength of 10 mT that was considerably higher than in many published reports (e.g. in [31]: 1 mT at 50 Hz). We also sought to combine both overlapping harmonic ranges by sequentially exposing our MNs first to 10 Hz followed by 2 Hz. To this end, we established in preliminary trials that a sequence of three 10 Hz treatments followed by one final 2 Hz treatment (see Material and Methods) was optimal, whereas, e.g., 10 Hz two times with 2 Hz one time or 10 Hz one time and 2 Hz three times or 10 Hz three times or 2 Hz three times was less efficient. In essence, the one with three times at 10 Hz and one time at 2 Hz seemed to provide most promising results and were thus selected for systematic experiments with more cell lines. We revised Results, section 3.2. (line 558 and following) accordingly.

Of note, our study is more or less a proof of concept study on this kind of magnetic stimulation of patient-derived motoneurons including ALS patients, thus it is more concept- and hypothesis-generating. Therefore, follow-up studies are clearly warranted to (i) reproduce the results in more cell lines (with different mutations, from other diseases), (ii) to further „titrate“ the most efficient frequencies/stimulation conditions, (iii) to increase our understanding of the cellular responses to the different magnetic simuli and (iv) also to further understand the underlying mechanisms. Thus we did also include a “limitaton of the study“ and “future outlook” paragraph in the end of the discussion (see also reviewer #1&3).

5) Rescue of axonal regeneration defects in FUS-ALS through MS:How does regrowth after cutting the axons relate to disease changes in ALS? The axons are not cut in the disease…

Response. The reviewer is right that axons are not cut in ALS. There were two reasons why we used this setup that enabled us two address two questions. First, we wanted to analyse axonal growth in WT vs. mutants and whether this can be beneficially affected by MS. Since MNs need first to differentiate and mature properly (see discussion above), we had to find a way of being able to analyse growth cones after at least 30 DIV. This is not possible in an uncompartmentalized culture format, since after >30 DIV one has a highly established neurite network in which single axonal growth cones cannot be identified in the dense soma-dendritic meshwork. We thus decided to use MFCs. But also in these it takes significant time until the axons grow through the microchannels, and there are significant inter-channel variations. Thus, we needed to have a setup in which we can let the MNs mature properly and then create a defined starting point of axonal growth cones. This let us decide for the mechanical axonotomy model at the distal exit side of the MFCs, which is known to only show axonal outgrowth and, using 900µm channels, these were purely MNs (PMID: 32151030, PMID: 29362359).

The second reason was that we intended to use this axonotomy model to investigate whether FUS-ALS mutant MNs are impaired in the axonal re-growth and whether this might be influenced beneficially by MS. This is an important question also in ALS patients, since there is a debate about axonal sprouting and remodelling of the neuromuscular unit (i.e. chronic denervation revealed in electropmyography recordings). Furthermore, since neurodegeneration in ALS is supposed to be a “dying back” event (see also PMID: 29362359, here Fig. 2 for FUS-ALS), with the first steps of neurodegeneration happening at the distal axon, any kind of improved axonal sprouting might induce functional recovery. These are the reasons why we utilized this setup here.

6) Please speculate why 10/2 Hz shows the best organelle motility recovery (but not 10 Hz or 2 Hz alone), but 10 Hz only impacted cone regrowth.

Response: We appreciate this comment. We believe that the current take-home message from the entire Z-score profile analysis (table 2, novel supplement tables) shows that higher frequencies (10 and 10/2 Hz) are more efficient as compared to 2 Hz only (Table 2). We would like to point out that our single parameter analysis might be missleading in that and want to point towards the whole multi-parameter profiles. We considered the whole profile analysis as more comprehensive and conclusive and decided to proceed with higher frequencies only (i.e. 10 Hz). We tried to make this more clear in the revised version (Results, section 3.2., line 725 and following) but also tried to point out the yet existing limitations of our initial study (Discussion, line 1068 and following) and want thus not too much stress these differences prior additional and independent follow-up studies.

7) Figure 4: Were more images analyzed or only those shown in the figure? Did all images show this same phenomenon? I would like to see at least a basic quantification here, since this figure seems to be the basis for the “therapeutic potential” which is highly emphasized in the title, abstract, and conclusion of the paper. At minimum, how many images clearly showed the illustrated changes vs how many showed no changes vs how many showed alternative changes.

Response: We totally agree with the reviewer and have to say that we intensively discussed this part prior submission and once more during revisions. The experiments were technically demanding and there had been many experiments in which cells were lost or significantly altered during the prosessing for the EM. This is the reason why we only obtained reliable results from 1 control and 1 mutant cell line. We were and are still also concerned about quantifying the images, since there are some limitation to investigate the peripheral mitochondria inside axons using the presented technology, which are extremely fragile and one never can guarantee how many are lost during sample processing. Therefore, it was difficult to get enough mitochondria to quantify and thus we wanted to avoid any misjudgements. Along with the reviewer’s sorrow we decided to relocate these results to the supplements (now figure S7) clearly stating that these are very preliminary to avoid any kind of overemphasizing.

8) Please provide a citation or other justification for using fold-change in figures 5-7. How are the raw intensity values measured for each image?

Response: We believed that fold-changes make it easier to catch a change because the reader does not have to compare two bars any more. However, we replaced the fold-change analysis in figure 5 and 7 by dot plots, consistent to figure 4 and 6. We hope that this harmonization avoids any irritation now. Moreover, revised M&M, section 2.7. provides details about the IF quantifications.

9) Statistics: Supplemental data with detailed results of all statistical comparisons should be provided.

Response: We added a new Table S1 to the supplemental part providing details about the ANOVA of figure 2. Moreover, we carefully revised the significance indicators on top of all quantifications (showing asterisks and “ns”) for a better logic and consistent pattern. More statistical comparisons are enabled now through the figures in the supplemental part showing the individual lines corresponding to the pooled analyses in the main manuscript. Since many analyses yielded negative data (i.e. no phenotype, no significant change) it was often sufficient to add the following statement to the figure legend: “One-way ANOVA with Bonferroni post-hoc test revealed no significant alteration in any pairwise comparison.” Thus, it was not necessary to prepare additional tables with all statistical comparisons.

10) “Ditto” is a very informal word and should be replaced with “likewise” or something similar.

Response: We did so.

Reviewer 3 Report

In this manuscript, Kandhavivorn et al. use iPSC-derived motor neurons with different ALS-associated FUS-mutations to assess phenotypes of axonal organelle motility and regeneration and evaluate the potential of magnetic field stimulation to modify these phenotypes. The authors report that FUS mutant lines, compared with healthy and isogenic control lines, are deficient in axonal trafficking of mitochondria and lysosomes, have reduced axonal regenerative sprouting after axotomy, altered mitochondrial morphology, and modulated cytoskeletal integrity. In addition, Kandhavivorn et al. report that administration of low frequency alternating current magnetic fields in culture rescues these phenotypes in culture.

General comments:

Although not completely novel as a concept (e.g., PMID: 36359255), the manuscript by Kandhavivorn et al. employs an interesting approach, evaluating the effect of magnetic stimulation on cultured FUS mutant iPSC-derived motor neurons. This approach is of interest to the field due to its translational potential and fits within the remit of this special issue on ALS in Cells. Overall, the manuscript is well-structured and mostly clear but would benefit from proof-reading/editing by a native speaker to improve the overall clarity. However, there are several major issues that the authors need to address (see also specific comments below):

Magnetic stimulation (MS) is not routinely used in the iPSC field or with neuronal cultures; a thorough evaluation of its effect on neuronal physiology in culture therefore seems warranted and would strengthen the manuscript. One could imagine that MS elicits its effects by modulating neuronal ion channels – do the authors have data on whether MS modulates neuronal activity, for instance, using patch-clamping or multi-electrode array recordings?

The differentiation efficiency of the motor neuron differentiation protocol is comparatively low, generating only ~50-70% neurons (MAP2/SMI32/TUJ1-positivity); more specific motor neuron markers such as ISLET1 or ChAT have not been assessed. Effects of magnetic stimulation could therefore be mediated by non-neuronal cells in culture rather than (motor) neuronal cells. For instance, do the authors observe GFAP-positive astrocytes in their cultures? Could they be affected by MS?

Furthermore, the authors tend to overstate some of their conclusions. This should be rectified and toned down. For example, “our study thus points towards an exciting novel treatment option in ALS” (line 48); see the specific comments, for more examples. 

Specific comments:

-       Methods:

o   “four cell lines from healthy volunteers” (l. 114) – technically, these are three healthy volunteers and one CRISPR-generated isogenic line;

o    in light of a recent paper (PMID: 36764301), it is commendable that the authors used a gender-matched approach between controls and disease condition – this should be mentioned

o   Table 1: Is FUS P252L really an isogenic “control”?

o   Even though the MN differentiation protocol has been published before, the methods section should describe key steps of the differentiation protocol/key media constituents

o   Can the authors elaborate on the reasoning for the chosen stimulation paradigm and frequencies?

o   “To quantify the fluorescent intensity of neuronal and motoneuronal markers and others (see above), their mean intensities were analyzed in FIJI using standard thresholding and masking tools for segmentation of the filamentous markers and normalized by total neurite length per image” (line 199) – this seems to contradict figure 1 where the authors show % positive cells. Please can the authors clarify?

o   “Trackers were added from a 1 mM stock in DMSO directly to culture supernatants and incubated for 1 hour at 37 °C.” (line 255) – to the soma or axonal side? 

-       Discussion:

o   “To this end, we newly established a peripheral axonal injury model through axotomy in microfluidic chambers in order to investigate advancing growth cones in MNs derived from healthy donors and FUS-ALS patients” (line 750) – models of axonal injury using microfluidics have been used several times before; the authors are urged not to overstate the novelty of their approach.

o   “Thus, the restoration of axonal transport defects through MS point to a powerful non-pharmacological, non-invasive therapy in ALS and presumably other (motor) neuropathies/spinal cord injury.” (line 778) – this seems like a substantial leap; the approach needs thorough validation etc, the authors are advised to tone down this statement.

o   A statement on limitations of this work could make for a more balanced discussion.

-     Conclusions:

o   “This work demonstrates a high therapeutic potential of magnetic field treatments on motoneuron disease, since it restored several hallmarks of ALS pathogenesis” (line 783).” Again, the authors are advised not to overstate the significance of their findings.

-       Figures:

-       Summary schematic:

o   suggests that the authors actually used a model system for the neuromuscular junction such as co-cultures of spinal motor neurons and myotubes (e.g., C2C12). As the authors merely analysed axons, this should be amended.

-       Figure 1: 

o   The authors should also plot individual data points (here and in the other figures) – this would allow the reader to get a better feel for the robustness of the data as well as the intrinsic variability of the iPSC differentiations/lines.

o   As the overall MN efficiency seems comparatively low, the authors should also assess the expression of additional, commonly used MN markers such as ISLET1 and ChAT and the expression of potential glial contaminants such as GFAP-positive astrocytes.

-       Figure 2: 

o   How do the authors explain that the combined 10/2 stimulation pattern seems more beneficial than 10 Hz only?

o   If the z-score represents the derivation of condition x from the untreated Ctr pool, why does the untreated Ctr pool also show a derivation (from itself)? Can the authors clarify?

o   g-i) it is slightly challenging to distinguish the different conditions as is, perhaps the authors could consider slightly changing the colour scheme/line pattern here.

-       Figure 3:

o   It would be interesting to see images of axonal outgrowth before axotomy – do the mutant cells show a growth phenotype compared with control?

o   Based on the images in b), it also looks like the number of regenerating fibres is increased after MS in the control group. Could the authors quantify this?

o   The analysis of mean speed of outgrowth in c) seems perhaps slightly unnecessarily complicated – could the authors also provide a quantification of the absolute outgrowth in um per genotype?

o   Axotomised axons usually show signs of fragmentation or formation of degeneration bulbs – do the authors observe such a phenotype, and if yes, does their treatment have a beneficial effect?

-       Figure 5:

o   The analysis of cleaved caspase 3 (CC3) intensity is not convincing. The images suggest that CC3 is particularly expressed in MAP2-negative (likely non-neuronal) cells. To limit their quantification to neurons and therefore make claims on neuronal apoptosis, the authors should analyse the % CC3+ in MAP2+ or TUJ1+ cells in mutant and control lines.

Author Response

Dear Editor, dear reviewers,

we are very grateful for the many constructive suggestions raised by the three reviewers. As you can find below, we tried hard to adress and answer every single comment. We believe that this significantly strengthens our manuscript and hope that it is now acceptable for publication.

Comments and Suggestions for Authors

In this manuscript, Kandhavivorn et al. use iPSC-derived motor neurons with different ALS-associated FUS-mutations to assess phenotypes of axonal organelle motility and regeneration and evaluate the potential of magnetic field stimulation to modify these phenotypes. The authors report that FUS mutant lines, compared with healthy and isogenic control lines, are deficient in axonal trafficking of mitochondria and lysosomes, have reduced axonal regenerative sprouting after axotomy, altered mitochondrial morphology, and modulated cytoskeletal integrity. In addition, Kandhavivorn et al. report that administration of low frequency alternating current magnetic fields in culture rescues these phenotypes in culture.

General comments:

Although not completely novel as a concept (e.g., PMID: 36359255), the manuscript by Kandhavivorn et al. employs an interesting approach, evaluating the effect of magnetic stimulation on cultured FUS mutant iPSC-derived motor neurons. This approach is of interest to the field due to its translational potential and fits within the remit of this special issue on ALS in Cells. Overall, the manuscript is well-structured and mostly clear but would benefit from proof-reading/editing by a native speaker to improve the overall clarity. However, there are several major issues that the authors need to address (see also specific comments below):

Response: We deeply appreciate the very positive feedback of reviewer#3. Of note, PMID: 36359255 investigated the effect of MS for deriving neuronal differentiation of murine cells while our study focusses on rescuing the defects of human motoneurons from ALS patients.

Magnetic stimulation (MS) is not routinely used in the iPSC field or with neuronal cultures; a thorough evaluation of its effect on neuronal physiology in culture therefore seems warranted and would strengthen the manuscript. One could imagine that MS elicits its effects by modulating neuronal ion channels – do the authors have data on whether MS modulates neuronal activity, for instance, using patch-clamping or multi-electrode array recordings?

Response: Our study is an early proof of concept study of this kind of magnetic stimulation of patient-derived motoneurons including ALS patients, thus more concept- and hypothesis-generating. Thus, as the reviewer wrote, there is a clear need for follow-up studies to (i) validate the results in more ALS cell lines bearing, e.g., mutant SOD1, C9orf72, TDP43 and other disease models (HD, AD, PD, etc.), (ii) to further „titrate“ the most efficient frequencies/stimulation conditions, (iii) to increase our understanding of the cellular responses to the different magnetic simuli and (iv) also to further understand the underlying mechanisms. These also include such analyses the reviewer suggests. But these would also (at least partially) need different set ups to be able to measure e.g. Ca2+ imaging or patch clamb during stimulation, since these effects might be transient. Also MEA recording would be interesting, however our setup dissabled the use of MEA chambers in the magnetic stimulator. Thus, we believe that these analyses are beyond the scope of your manuscript here, but of course would be very interesting and are indeed needed to further understand the mechanisms of MS on human neurons. We included a comment on this in the novel limitation and future outlook paragraph at the end of the discussion.

The differentiation efficiency of the motor neuron differentiation protocol is comparatively low, generating only ~50-70% neurons (MAP2/SMI32/TUJ1-positivity); more specific motor neuron markers such as ISLET1 or ChAT have not been assessed. Effects of magnetic stimulation could therefore be mediated by non-neuronal cells in culture rather than (motor) neuronal cells. For instance, do the authors observe GFAP-positive astrocytes in their cultures? Could they be affected by MS?

Response: The effect on non-neuronal cells including possible indirect effects of MS mediated via non-neuronal cells is interesting. Of course, GFAP-positive astrocytes could be affected by MS, but our protocol does not yield any GFAP-positive astrocytes nor other non-neuronal cells except of undifferentiated NPCs. We used a standard differentiation protocol which we have used throughout the last years and we showed in detail in our previous publications that we derive only neurons and NPCs out of this protocol, but no other neuroectodermal or non neuroectodermal cells (e.g. PMID: 29362359, PMID: 26946488, PMID: 31108504). We also show that neurons develop normally but then aquire functional deficits (starting at DIV21) followed by structural axonal degeneration (DIV60) and cell death (DIV 110), see Naumann et al for details (PMID: 29362359). We also did show in our previous publications that our MNs (WT and Mut) are already fully matured at the timepoints investigated in the current study and do express more specific motor neuron markers such as Hb9, Islet and ChAT at 30-60 DIV in Mut vs WT, (Naumann et al., 2018 - PMID: 29362359, Naujock et al., 2016 - PMID: 26946488, Bursch et al, 2019 - PMID: 31108504). While we do think that the differentiation yield is not bad, all our analysis are cell based, i.e., clearly only in neurons (e.g. by MAP staining and normalization). Thus, we believe that we indeed depict results of (motor)neurons. The only non-neuronal effect could come only from remaining NPCs.

Furthermore, the authors tend to overstate some of their conclusions. This should be rectified and toned down. For example, “our study thus points towards an exciting novel treatment option in ALS” (line 48); see the specific comments, for more examples.

Response: We toned down throughout the manuscript and depict putative limitations more clearly in the revised version of the manuscript.

Specific comments:

-Methods:

-“four cell lines from healthy volunteers” (l. 114) – technically, these are three healthy volunteers and one CRISPR-generated isogenic line;

Response: We corrected this.

-in light of a recent paper (PMID: 36764301), it is commendable that the authors used a gender-matched approach between controls and disease condition – this should be mentioned

Response: We actually did, we used a young male matched to the young male 495x but mid age female to match the mid age female mutants!

-Table 1: Is FUS P525L really an isogenic “control”?

Response: We did rename it properly.

-Even though the MN differentiation protocol has been published before, the methods section should describe key steps of the differentiation protocol/key media constituents

Response: We did so.

-Can the authors elaborate on the reasoning for the chosen stimulation paradigm and frequencies?

Response: In our search for a meaningful frequency setting for our MS experiments, we encountered a growing number of published reports about benefical effects of extremely low frequency electromagnetic field (ELF-EMF) applications in the range of 1-100 Hz. In essence, it was difficult for us to decide which of the already tested frequencies could serve as the most promising and meaningful starting point for our own MS experiments on human iPSC-derived MNs. This was simply due to the multitude of different disease models and biological readout assays tested with unclear relevance for our own model system and biological question we wished to address. Thus, in order to streamline our proof of concept study and to avoid tedious screenings to empirically establish efficient frequency settings, we decided to use AC rectangular instead of sine waves. Unlike the homogenous sine waves, rectangular waves have the benefit of producing a spectrum of higher harmonic frequencies according to the Fourier analysis (see also Material and Methods), thereby allowing to test multiple frequencies simultaneously. For example, a basal frequency of 2 Hz rectangular waves results in 2 Hz of amplitude 1 + 6 Hz of amplitude 1/3 + 10 Hz of amplitude 1/5 + 14 Hz of amplitude 1/7 + etc. Likewise, a basal frequency of 10 Hz results in 10 Hz of amplitude 1 + 30 Hz of amplitude 1/3 + 50 Hz of amplitude 1/5 + 70 Hz of amplitude 1/7 + etc. We decided to test 2 Hz to capture the low ELF-EMF range and 10 Hz for the higher range. To counterbalance the decreasing amplitude at higher harmonic frequencies, we utilized a relatively high field strength of 10 mT that was considerably higher than in many published reports (e.g. in [31]: 1 mT at 50 Hz). We also sought to combine both overlapping harmonic ranges by sequentially exposing our MNs first to 10 Hz followed by 2 Hz. To this end, we established in preliminary trials that a sequence of three 10 Hz treatments followed by one final 2 Hz treatment (see Material and Methods) was optimal, whereas, e.g., 10 Hz two times with 2 Hz one time or 10 Hz one time and 2 Hz three times or 10 Hz three times or 2 Hz three times was less efficient. In essence, the one with three times at 10 Hz and one time at 2 Hz seemed to provide most promising results and were thus selected for systematic experiments with more cell lines. We revised Results, section 3.2. (line 574 and following) accordingly.

Of note, our study is more or less a proof of concept study on this kind of magnetic stimulation of patient-derived motoneurons including ALS patients, thus it is more concept- and hypothesis-generating. Therefore, follow-up studies are clearly warranted to (i) reproduce the results in more cell lines (with different mutations, from other diseases), (ii) to further „titrate“ the most efficient frequencies/stimulation conditions, (iii) to increase our understanding of the cellular responses to the different magnetic simuli and (iv) also to further understand the underlying mechanisms. Thus we did also include a “limitaton of the study“ and “future outlook” paragraph in the end of the discussion (see also reviewer #1&3).

-“To quantify the fluorescent intensity of neuronal and motoneuronal markers and others (see above), their mean intensities were analyzed in FIJI using standard thresholding and masking tools for segmentation of the filamentous markers and normalized by total neurite length per image” (line 199) – this seems to contradict figure 1 where the authors show % positive cells. Please can the authors clarify?

Response: In figure 1 we did not measure fluorescence intensities but only counted marker-positive cells per total cells after an unbiased thresholding. We tried to make this clearer in the revised M&M, section 2.7.

-“Trackers were added from a 1 mM stock in DMSO directly to culture supernatants and incubated for 1 hour at 37 °C.” (line 255) – to the soma or axonal side?

Response: To both sides, we added this info to M&M, section 2.9.

-Discussion:

-“To this end, we newly established a peripheral axonal injury model through axotomy in microfluidic chambers in order to investigate advancing growth cones in MNs derived from healthy donors and FUS-ALS patients” (line 750) – models of axonal injury using microfluidics have been used several times before; the authors are urged not to overstate the novelty of their approach.

Response: We toned down

-“Thus, the restoration of axonal transport defects through MS point to a powerful non-pharmacological, non-invasive therapy in ALS and presumably other (motor) neuropathies/spinal cord injury.” (line 778) – this seems like a substantial leap; the approach needs thorough validation etc, the authors are advised to tone down this statement.

Response: We toned down.

-A statement on limitations of this work could make for a more balanced discussion.

Response: We did include a new paragraph about the limitations of the current report and a future outlook to the end of the discussion (line 1068 and following).

-Conclusions:

-“This work demonstrates a high therapeutic potential of magnetic field treatments on motoneuron disease, since it restored several hallmarks of ALS pathogenesis” (line 783).” Again, the authors are advised not to overstate the significance of their findings.

Response: We toned down.

-Figures:

-Summary schematic:

-suggests that the authors actually used a model system for the neuromuscular junction such as co-cultures of spinal motor neurons and myotubes (e.g., C2C12). As the authors merely analysed axons, this should be amended.

Response: We removed the muscle and the neuromuscular junction to avoid this misunderatanding, but still show the distal axon as our experiments address critical events at the distal axon.

-Figure 1: 

-The authors should also plot individual data points (here and in the other figures) – this would allow the reader to get a better feel for the robustness of the data as well as the intrinsic variability of the iPSC differentiations/lines.

Response: We provide scatter dot plots for figure 4, 5, 6, 7 in the revised version now. For the box plots in figure 2, we disfavor scatter dot plots as the resultant scatter clouds would be too dense (i.e. thousands of individual organelle tracks). The outliers beyond the whiskers are already hard to discern. We also prefer to have bar graphs in figure 1 as a preferable way of abstraction that facilitates to convey the key message from so many bar graphs at the same time as compared to the more complex scatter clouds. As for comprehensiveness and transparency, we added the sample size n and further statistical details to each figure legend. Please do also refer to the suppl. figures in which we show the corresponding results of the individual lines to allow the reader to better judge on the variability between cell lines.

-As the overall MN efficiency seems comparatively low, the authors should also assess the expression of additional, commonly used MN markers such as ISLET1 and ChAT and the expression of potential glial contaminants such as GFAP-positive astrocytes.

Response: As mentioned above and for the other reviewers, our protocol does not yield any GFAP-positive astrocytes nor other non-neuronal cells despite undifferentiated NPCs. We used a standard differentiation protocol which we have been using throughout the last years and we showed in detail in our previous publications that we derive only neurons and NPCs out of this protocol, but no other neuroectodermal or non neuroectodermal cells (e.g. PMID: 29362359, PMID: 26946488, PMID: 31108504). We also showed that neurons develop normally but then acquire functional deficits (starting at DIV21) followed by structural axonal degeneration (DIV60) and cell death (DIV 110), see Naumann et al for details (PMID: 29362359). We also did show in our previous publications that our MNs (WT and Mut) are already fully matured at the timepoints investigated in the current study and do express more specific motor neuron markers such as Hb9, Islet and ChAT at 30-60 DIV in Mut vs WT, (Naumann et al., 2018 - PMID: 29362359, Naujock et al., 2016 - PMID: 26946488, Bursch et al, 2019 - PMID: 31108504). While we do think that the differentiation yield is not bad, all our analysis are cell-based, thereby addressing only neurons (e.g. by MAP staining and normalization). Thus, we believe that we indeed depict results of (motor)neurons. The only non-neuronal effect could come only from remaining NPCs.

-Figure 2:

-How do the authors explain that the combined 10/2 stimulation pattern seems more beneficial than 10 Hz only?

Response: We believe that the current take-home message from the entire Z-score profile analysis (table 2, novel supplement tables) shows that higher frequencies (10 and 10/2 Hz) are more efficient as compared to 2 Hz only (Table 2). We would like to point out that our single parameter analysis might be missleading in that and want to point towards the whole multi-parameter profiles. We considered the whole profile analysis as more comprehensive and conclusive and decided to proceed with higher frequencies only (i.e. 10 Hz). We tried to make this more clear in the revised version (Results, section 3.2., line 725 and following) but also tried to point out the yet existing limitations of our initial study (Discussion, line 1068 and following) and want thus not too much stress these differences prior additional and independent follow-up studies.

-If the z-score represents the derivation of condition x from the untreated Ctr pool, why does the untreated Ctr pool also show a derivation (from itself)? Can the authors clarify?

Response: All Z-scores are normalized to the Ctrl untreated PROXIMAL condition, therefore the Ctrl untreated DISTAL part already deviates from zero. We amended M&M, section 2.11., accordingly.

-g-i) it is slightly challenging to distinguish the different conditions as is, perhaps the authors could consider slightly changing the colour scheme/line pattern here.

Response: We rearranged figure 2 completely, see also our detailed response about this point for reviewer#1.

-Figure 3:

-It would be interesting to see images of axonal outgrowth before axotomy – do the mutant cells show a growth phenotype compared with control?

Response: There were two reasons why we used this setup that enabled us two address two questions. First, we wanted to analyse axonal growth in WT vs. mutants and whether this can be beneficially affected by MS. Since MNs need first to differentiate and mature properly (see discussion above), we had to find a way of being able to analyse growth cones after at least 30 DIV. This is not possible in an uncompartmentalized culture format, since after >30 DIV one has a highly established neurite network in which single axonal growth cones cannot be identified in the dense soma-dendritic meshwork. We thus decided to use MFCs. But also in these it takes significant time until the axons grow through the microchannels, and there are significant inter-channel variations. Thus, we needed to have a setup in which we can let the MNs mature properly and then create a defined starting point of axonal growth cones. This let us decide for the mechanical axonotomy model at the distal exit side of the MFCs, which is known to only show axonal outgrowth and, using 900µm channels, these were purely MNs (PMID: 32151030, PMID: 29362359).

The second reason was that we intended to use this axonotomy model to investigate whether FUS-ALS mutant MNs are impaired in the axonal re-growth and whether this might be influenced beneficially by MS. This is an important question also in ALS patients, since there is a debate about axonal sprouting and remodelling of the neuromuscular unit (i.e. chronic denervation revealed in electropmyography recordings). Furthermore, since neurodegeneration in ALS is supposed to be a “dying back” event (see also PMID: 29362359, here Fig. 2 for FUS-ALS), with the first steps of neurodegeneration happening at the distal axon, any kind of improved axonal sprouting might induce functional recovery. These are the reasons why we utilized this setup here.

-Based on the images in b), it also looks like the number of regenerating fibres is increased after MS in the control group. Could the authors quantify this?

Response: We had to choose ROI (region of interest) in the sense to be able to measure outgrowth behaviour best, which does not necessarily depict representative amounts of axons per se. Therefore, we exchanged the images to have better representative examples in the revised figure 4b (formerly figure 3). We provide now overview images for each condition to better illustrate similar axon density in the newly populated distal areas after axotomy whereas the inlets show magnifications to illustrate the difference in axon outgrowth and its rescue through MS.

-The analysis of mean speed of outgrowth in c) seems perhaps slightly unnecessarily complicated – could the authors also provide a quantification of the absolute outgrowth in um per genotype?

Response:

We appreciate the suggestion of investigating the absolute outgrowth. In the initial experimental design we deliberately focused on the speed as a major readout, because it is unaffected by the length of each video and appears to be less error prone than the raw outgrowth in µm. Since we cannot predict the exact behaviour of the axotomy – whether it will be a clean “cut” at the exit site, or a shearing and potential pulling out of distal parts of the axon, we deemed the absolute outgrowth as too inaccurate. With the available videos we cannot analyze the absolute outgrowth since most of the videos don’t contain the silicone edge as a reference point. We are therefore afraid, that this analysis, albeit interesting, is beyond the scope of the current revision process since we would have to redo all MS experiments in MFCs.

-Axotomised axons usually show signs of fragmentation or formation of degeneration bulbs – do the authors observe such a phenotype, and if yes, does their treatment have a beneficial effect?

Response: The reviewer is right, axotomised axons might show signs of degeneration bulbs and so on. We tried to get a sharp truncation boundary at the distal channel exit using the mechanical axotomy described here. Our mechanical shearing flow protocol always leaves the channel exit area (where the subsequent imaging is then performed) blank with the remaining truncated axon trunks inside channels reaching very close to their distal end. Thus, if present, the degeneration bulbs would be inside the channels, and thus evaded further analyis.

-Figure 5:

-The analysis of cleaved caspase 3 (CC3) intensity is not convincing. The images suggest that CC3 is particularly expressed in MAP2-negative (likely non-neuronal) cells. To limit their quantification to neurons and therefore make claims on neuronal apoptosis, the authors should analyse the % CC3+ in MAP2+ or TUJ1+ cells in mutant and control lines.

Response: Indeed we already showed %CC3+ in MAP2+ neurons and explain our quantification method with better clarity in the revised M&M, section 2.7. We also spotted that we accidently swapped some images what might have caused some misunderstandings in the previous version. We corrected this in the revised version.

Round 2

Reviewer 1 Report

I thank the authors for taking my comments and suggestions critically and professionally. While they have clarified many of the comments raised in the manuscript review, there are still critical points that need to be resolved.

Results

3.1 Neuronal characterization and the effect of magnetic stimulations on neuronal differentiation.

Minor points

2) Show in the supplementary figures representative images of the control and mutant about the markers used in the analysis.

Response: We added to the supplemental part a new figure S1 to provide more representative examples from cell lines used for the pooled analysis in the corresponding figure 1b-f. Likewise, we added a new figure S3 to provide more representative examples from cell lines used for the pooled analysis in the corresponding figure 1g-j.

The authors show more images used to perform the analysis in section 3.1. The appearance of TUJ1 is not great and is also different from what the authors show in the main figure. The cells appear to be stressed. How do the authors justify these differences and their quantification?

3.3 Rescue of axonal regeneration defects in FUS-ALS through MS.  

Major points

1) The axotomy performed in the axonal chamber is mechanical. The authors claim to have committed an axotomy using the pipette in the materials and methods paragraphs. A mechanical axotomy is less accurate than a chemical one (e.g. trypsin). This often makes the axotomy incomplete, with possible truncated cuts on the axons but not all at the same height of the microfluidics (to be clear, at the level of the microgrooves) that the authors showed in the cartoon. It would be worth performing chemical damage. 

Response: On a side note: due to the insertion of the new figure 3 (HC profiles), the axotomy data are now in figure 4. The aims of the experiment were to compare the ability of axonal regeneration in healthy versus FUS-ALS patients‘ neurons and whether the effect of magnetic stimulations can rescue the defect of regeneration after the injury. The reviewer is correct that we used a mechanical way to perform the axotomy. Even in the case it would yield an incomplete axotomy, this does not affect our quantifications since we quantified the regrowth in a defined time from a defined starting position to a defined end position.

The authors claim to have quantified the growth in a defined time from a defined initial position to a defined final position. For this reason, it is impossible to analyse the axonal growth of the MNs after injury, only the growth rate after rescue. They perform the analyses 24 hours after treatment. Indeed the images shown (initial position) do not show any difference in the neurons of healthy patients compared to those of FUS-ALS patients and with or without magnetic stimulation. Authors should emphasize this aspect.

NB There are several published papers that have shown faster growth after axotomy in ALS iPSC-derived MNs compared to its isogenic control line.

In contrast, we believe that chemical axotomy is not feasable in the MFC setup: We doubt that an enzymatic axotomy would result in a sharp truncation boundary at the distal channel exit at all because, e.g., trypsin will penetrate the microchannels during its incubation time to various extents depending on the local microflow, axon bundle density, obstacles etc. Therefore. each channel could potentially have a different truncation boundary. Moreover, the trypsin treatment would probably digest the crucial laminin coating in our MFCs, thereby hampering any new outgrowth. For these reasons, we disfavour proteolytic approaches. Instead, our mechanical shearing flow protocol always leaves the channel exit area (where the subsequent imaging is then performed) blank with the remaining truncated axon trunks inside channels reaching very close to their distal end as we verified by careful viewing right after the axotomy. We revised the discussion to make this point accordingly (line 952).

There are already published papers on hiPSC-derived MNs and primary mouse MNs using enzymatic approaches. This strategy is possible with the right methodological precautions.

3.4 Recovery of compromised mitochondria morphology in FUS-ALS after MS.  

1) How many times was the experiment carried out, and on how many lines?

2) Would it be possible to quantify this evidence? How many mitochondria show an aberrant phenotype from the total number of mitochondria analysed?

Response: We totally agree with the reviewer and have to say that we intensively discussed this part prior submission and once more during revisions. The experiments were technically demanding and there had been many experiments in which cells were lost or significantly altered during the prosessing for the EM. This is the reason why we only obtained reliable results from 1 control and 1 mutant cell line. We were and are still also concerned about quantifying the images, since there are some limitation to investigate the peripheral mitochondria inside axons using the presented technology, which are extremely fragile and one never can guarantee how many are lost during sample processing. Therefore, it was difficult to get enough mitochondria to quantify and thus we wanted to avoid any misjudgements. Along with the reviewer’s sorrow we decided to relocate these results to the supplements (now figure S7) clearly stating that these are very preliminary to avoid any kind of overemphasizing.

After this statement, the data relating to these experiments should not be published in the leading figures, much less in the supplementary figures.

3.6 Magnetic stimulations modulated cytoskeleton integrity in MNs with FUS mutations. 

Major points

2) Assuming the image of Figure 6 was representative. The authors state a lower expression of MAP2 in the FUS mutant and that MS treatment restores the physiological condition. Why is it impossible to appreciate the same phenotype in the image figure 5b?

Response: Our apologies for having not perfectly representative images so far. We replaced them in the revised version of the manuscript.

The authors do not show the same difference in MAP2 signal intensity in Figures 5 and 6

Author Response

Comments and Suggestions for Authors

I thank the authors for taking my comments and suggestions critically and professionally. While they have clarified many of the comments raised in the manuscript review, there are still critical points that need to be resolved.

Response: We thank the reviewer for the words of praise for our efforts. We also appreciate the very constructive way of revision and tried hard to address all remaining issues.

Results

3.1 Neuronal characterization and the effect of magnetic stimulations on neuronal differentiation.

Minor points

The authors show more images used to perform the analysis in section 3.1. The appearance of TUJ1 is not great and is also different from what the authors show in the main figure. The cells appear to be stressed. How do the authors justify these differences and their quantification?

Response: We wish to thank the reviewer for having keenly spotted the discrepancy and carefully re-inspected the images of question. We found that the majority of our TUJ1 immunostainings looks indeed fragmented regardless of genotype, treatment and batch. In light of the smooth, continuous appearance of all other cytoskeletal markers used in this study (SMI32, MAP2, a-tublulin, acetylated tubulin), we rejected a truly fragmented microtubule network due to extended ageing at 48 DIV as possible underlying cause. Instead, we consider a systematic technical problem (e.g. with the antibody batch) in the staining procedure as a more likely explanation. Therefore, to avoid irritations and possible erratic quantifications, we decided to remove all TUJ1 data from the manuscript (i.e. in Fig. 1 and S1, S2, S4), as we believe that the remaining SMI32 and MAP2 data are of high quality to still enable a faithful discrimination between general neuronal versus specific motoneuronal differentiation. We hope that the reviewer agrees that this measure is appropriate in order to streamline and consolidate the manuscript.

3.3 Rescue of axonal regeneration defects in FUS-ALS through MS.  

Major points

The authors claim to have quantified the growth in a defined time from a defined initial position to a defined final position. For this reason, it is impossible to analyse the axonal growth of the MNs after injury, only the growth rate after rescue. They perform the analyses 24 hours after treatment. Indeed the images shown (initial position) do not show any difference in the neurons of healthy patients compared to those of FUS-ALS patients and with or without magnetic stimulation. Authors should emphasize this aspect.

Response: The images of question in Fig. 4b show indeed the re-population of the distal assay area that was left blank right after the axotomy. The perceived density of the new axons here is a bit subjective but we agree with the reviewer that, due to the relatively subtle reduction of the growth cone speed in the untreated FUS mutant, it is hardly possible to spot any difference after just 24 hrs. Of note, however, is that we did not systematically investigated this in the presented setup since we had to select for traceable growth cones which might have biased such an analysis. We make this point now at the end of Results, section 3.3. Moreover, we revised the mid part of section 3.3. and M&M, section 2.11, and the discussion once more extensively to emphasize even more that (i) axons have progressively sprouted out from the distal channel exits in MFCs at 48 DIV before the axotomy was performed in all lines, (ii) that our axotomy created an entirely blank area to assay specifically the subsequent regenerative re-outgrowth (as opposed to general outgrowth in the course of regular differentiation) and (iii) that our mechanical truncation method is valid also considering several existing publications. Moreover, we have added a new Fig. S7 to the supplement to document the success of our complete axotomy with images from before and after the procedure.

NB There are several published papers that have shown faster growth after axotomy in ALS iPSC-derived MNs compared to its isogenic control line.

Response: We appreciate the reviewer’s point that we have to discuss our data more in light of conflicting other reports. We found that particularly Garone et al. 2021 (https://doi.org/10.1038/s42003-021-02538-8) found the opposite in what looks like a similar ALS-FUS model system (i.e. hiPSC-derived spinal MNs with a P525L mutation). We have added a new paragraph to our revised discussion to illuminate on possible reasons for the drastic, opposing discrepancy between our and their results. In essence, the reasons remain elusive but some feasible explanations are possible: different genetic backgrounds of the donors, different gene-editing technologies, different FUS-GFP-linkers, and, most importantly, different developmental stages when the axotomy was performed. Clearly, future studies are necessary for clarification.

There are already published papers on hiPSC-derived MNs and primary mouse MNs using enzymatic approaches. This strategy is possible with the right methodological precautions.

Response: We very much appreciate the reviewer’s expertise in enzymatic axotomy and will eagerly explore this interesting alternative in our future studies. Moreover, we were motivated by the reviewer’s comment to perform a further literature search and have found once more, to our reassurance, that mechanical axotomy in Xona MFC devices is a long standing, well-established standard method since at least 2005, even reported in Nature Methods amongst others (https://doi.org/10.1038/s41598-019-49214-w, https://doi.org/10.1038/nmeth777, https://doi.org/10.1371/journal.pone.0080722). Moreover, we have found in the abovementioned publication of Garone et al. 2021 (https://doi.org/10.1038/s42003-021-02538-8) a direct comparison of three different truncation methods in Fig. 4 herein. The authors directly compared axon regeneration after mechanical versus Trypsin- or Accutase-mediated axotomy and obtained amazingly consistent results in all three approaches (Fig. 4c, d, e herein). In essence, our mechanical approach was already validated by others. Therefore, we hope that the reviewer is now satisfied with our axotomy procedure, especially in light of the new Fig. S7 documenting the success of our truncation. Nevertheless we discussed the different ways of axonotomy in more detail in the revised version of the manuscript.

3.4 Recovery of compromised mitochondria morphology in FUS-ALS after MS.  

After this statement, the data relating to these experiments should not be published in the leading figures, much less in the supplementary figures.

Response: We have carefully re-considered our TEM data as we have taken the reviewer’s concern very serious. We have drawn the necessary consequences and have now removed the TEM data from the manuscript, as well as from the supplement and graphical abstract. We hope the reviewer agrees that this measure was appropriate in order to streamline and consolidate the manuscript.

3.6 Magnetic stimulations modulated cytoskeleton integrity in MNs with FUS mutations. 

The authors do not show the same difference in MAP2 signal intensity in Figures 5 and 6

Response: We kindly ask the reviewer to compare in the revised Fig.5 the MAP2 image for the FUS P525L mutant untreated in panel b against WT untreated in panel a (in pink). The neuritic staining intensity in the mutant is quite clearly lower. The perceived difference might be not as profound as in Fig. 6. In our opinion, this is, however, not a point of concern for three reasons:

  1. The MAP2 examples in Fig.5 were not primarily chosen to illustrate a MAP2 phenotype but simply follow a meaningful selection for CC3-stainings.
  2. As the quantification of MAP2 in Fig.6c (scatter dot plots) clearly shows, there is a natural variability within each data set. Thus, even when avoiding outliers, one will always select examples of different MAP2 intensity levels.
  3. Judging the MAP2 intensity levels just by viewing is subjective and potentially misleading, that’s why we are doing a normalization of the raw intensities first (by total neurite length, see M&M) for an unbiased, objective quantification.

For these reasons, we don’t see the benefit to exchange the shown examples in Fig.5 again until the perceived MAP2 differences “perfectly” match to that one in Fig.6. On the contrary, we find it very reassuring that both examples in Fig. 5 and 6 show an apparent reduction in MAP2 intensity in untreated mutant FUS, albeit to different degrees, which faithfully mirrors the variability of the data sets in Fig. 6c.

Reviewer 2 Report

The authors have made changes in the manuscript to address many of my concerns. The significance of the study has been softened and the authors now describe this study as a proof-of-concept. These are good revisions. 

My one remaining big concern is that the title and abstract are still worded rather strongly… “reveal a novel therapeutic approach,” which is no longer reflected in their discussion/conclusions since the authors now accurately address the fact that this work is preliminary and while conceptually showing some therapeutic potential, needs further verification. Therefore, the title and abstract should reflect that as well.

Author Response

The authors have made changes in the manuscript to address many of my concerns. The significance of the study has been softened and the authors now describe this study as a proof-of-concept. These are good revisions. 

My one remaining big concern is that the title and abstract are still worded rather strongly… “reveal a novel therapeutic approach,” which is no longer reflected in their discussion/conclusions since the authors now accurately address the fact that this work is preliminary and while conceptually showing some therapeutic potential, needs further verification. Therefore, the title and abstract should reflect that as well.

Response: We apologize for having overlooked this in the first revision. We have now revised the title and last sentence of the abstract as well and hope that our statements are now consistently toned down throughout the whole manuscript.

Reviewer 3 Report

Overall, the authors have addressed several of my previous comments. However, I still have some issues with this manuscript that I feel should be addressed:

- It is good that the authors have attempted to tone down their claims. I would still suggest changing the tile of the manuscript, too, considering that - as the authors point out themselves - this is "an early proof of concept study"; similarly, it seems premature to talk of a "novel treatment option in ALS" in the abstract 

- I’m not convinced by the authors’ claim that they derive only neurons and NPCs using their iPSC differentiation protocol. If not GFAP-positive, their non-(moto)neuronal cells will likely include S100B-positive (flat) cells (PMID: 36764301), also considering that the authors do not seem to employ antimitotics. The authors should tone down claims of 100% differentiation efficiency into neurons and NPCs and consider further optimising their differentiation protocol for future studies. 

- The authors now frequently describe their MNs as “fully matured”. One of their previous papers that they cite (PMID: 26946488) shows ~25% ChAT positivity. The iPSC field as a whole faces issues with impure and immature cultures, and even though the authors mature their cells for some time, they will still in essence represent a fetal developmental stage. “Noteably, we favored a late end point in our MS protocol because this reflects the adult onset of ALS in our disease model better than a time point earlier during differentiation” (line 977) – again, the authors are advised not to overstate the validity of their method.

- The authors have not answered my question “-It would be interesting to see images of axonal outgrowth before axotomy – do the mutant cells show a growth phenotype compared with control?”. I still think it would be important to see images before axotomy. Their reply seems to indicate that they used the axotomy to “normalise” axonal outgrowth, but in fact what they are evaluating is a post-lesion regenerative/sprouting response, which his is not the same as axonal outgrowth. What they are using is an axonal lesion model, not a model for axonal outgrowth (this claim needs to be removed from the discussion etc). And for this, it is important to ensure that all axons for the different genotypes are equally affected by the lesion. Otherwise, any difference in “regenerative” outgrowth could merely be the result of an incomplete axotomy, as the authors state themselves (“Since we cannot predict the exact behaviour of the axotomy – whether it will be a clean “cut” at the exit site, or a shearing and potential pulling out of distal parts of the axon, we deemed the absolute outgrowth as too inaccurate”). If no images of the axotomy site prior to the lesion are available to confirm successful axotomy, the authors should probable consider removing these data.

- The authors claim to have analysed the percentage of cleaved caspase 3 positive neurons in their reply, but they still provide a quantification of CC3 fluorescence intensity, which I think is challenging considering the high degree of apparent non-specific staining (small dots). I think the authors should perform the analysis I suggested (quantify number of CC3 positive MAP2 positive neurons), as it will be more informative. Also, why did the authors observe an apoptotic phenotype in what seems to be the same iPSC lines before (PMID: 29362359)?

Author Response

Overall, the authors have addressed several of my previous comments. However, I still have some issues with this manuscript that I feel should be addressed:

- It is good that the authors have attempted to tone down their claims. I would still suggest changing the tile of the manuscript, too, considering that - as the authors point out themselves - this is "an early proof of concept study"; similarly, it seems premature to talk of a "novel treatment option in ALS" in the abstract 

Response: We appreciate this concern that was already voiced by reviewer 2. We have now revised the title and last sentence of the abstract as well and hope that our statements are now consistently toned down throughout the whole manuscript.

- I’m not convinced by the authors’ claim that they derive only neurons and NPCs using their iPSC differentiation protocol. If not GFAP-positive, their non-(moto)neuronal cells will likely include S100B-positive (flat) cells (PMID: 36764301), also considering that the authors do not seem to employ antimitotics. The authors should tone down claims of 100% differentiation efficiency into neurons and NPCs and consider further optimising their differentiation protocol for future studies. 

Response: See our response to the next point, as both points are quite similar.

- The authors now frequently describe their MNs as “fully matured”. One of their previous papers that they cite (PMID: 26946488) shows ~25% ChAT positivity. The iPSC field as a whole faces issues with impure and immature cultures, and even though the authors mature their cells for some time, they will still in essence represent a fetal developmental stage. “Noteably, we favored a late end point in our MS protocol because this reflects the adult onset of ALS in our disease model better than a time point earlier during differentiation” (line 977) – again, the authors are advised not to overstate the validity of their method.

Response: We appreciate and acknowledge the reviewer’s concern about the heterogeneity often found in iPSC-derived cultures. Thus, we have expanded the third paragraph in our Discussion to illuminate more on these circumstances and that these might compromise the conclusiveness of our results with respect to the sough-after adult onset ALS model. We make the point now that even at 48 DIV we cannot rule out the presence of still fetal intermediate stages in our MN cultures and non-neuronal by-products such as S100B-positive flat cells. However, we also make the point that due to the architecture of MFCs, we yield nearly pure SMI32-positive axons in the distal compartment, as we have previously shown (PMID: 29362359, Naumann et al. 2018, Fig. 2b herein). Because most assaying for this study was actually carried out at the distal MFC compartment, we are still confident to predominantly address matured MNs in our setup.

- The authors have not answered my question “-It would be interesting to see images of axonal outgrowth before axotomy – do the mutant cells show a growth phenotype compared with control?”. I still think it would be important to see images before axotomy. Their reply seems to indicate that they used the axotomy to “normalise” axonal outgrowth, but in fact what they are evaluating is a post-lesion regenerative/sprouting response, which his is not the same as axonal outgrowth. What they are using is an axonal lesion model, not a model for axonal outgrowth (this claim needs to be removed from the discussion etc). And for this, it is important to ensure that all axons for the different genotypes are equally affected by the lesion. Otherwise, any difference in “regenerative” outgrowth could merely be the result of an incomplete axotomy, as the authors state themselves (“Since we cannot predict the exact behaviour of the axotomy – whether it will be a clean “cut” at the exit site, or a shearing and potential pulling out of distal parts of the axon, we deemed the absolute outgrowth as too inaccurate”). If no images of the axotomy site prior to the lesion are available to confirm successful axotomy, the authors should probable consider removing these data.

Response: We wish to apologize that we have not fully addressed the reviewer’s concern in the first revision and are grateful for her/his detailed account on this critical issue that has helped us to better address and clarify it. After careful reconsideration we agree and acknowledge that our experimental setup was indeed specifically addressing axonal regeneration after lesion, i.e. the new re-outgrowth from the lesion site, but not regular axonal outgrowth as well. Therefore, we have revised the mid part of section 3.3. and M&M, section 2.11, and the discussion once more extensively to emphasize even more that (i) axons have progressively sprouted out from the distal channel exits in MFCs at 48 DIV before the axotomy was performed, (ii) that our axotomy created an entirely blank area to assay specifically the subsequent regenerative re-outgrowth (as opposed to general outgrowth in the course of regular differentiation) and (iii) that our mechanical truncation method is reasonable in light of existing publications. Moreover, we have added a new Fig. S7 to the supplement to document the success of our complete axotomy with images from before and after the procedure. We hope that these revisions can provide a satisfactory clarification about this point.

- The authors claim to have analysed the percentage of cleaved caspase 3 positive neurons in their reply, but they still provide a quantification of CC3 fluorescence intensity, which I think is challenging considering the high degree of apparent non-specific staining (small dots). I think the authors should perform the analysis I suggested (quantify number of CC3 positive MAP2 positive neurons), as it will be more informative. Also, why did the authors observe an apoptotic phenotype in what seems to be the same iPSC lines before (PMID: 29362359)?

Response: We apologize for having overlooked this outstanding issue. We re-analyzed the CC3-images to express apoptotic levels as percentage of CC3-positive neurons within the MAP2-positive population. To this end, we have used a manual counting method as described for Fig. 1 and revised M&M, section 2.7. This re-analysis has further enhanced the trend to more apoptosis in the untreated mutant FUS pool (revised Fig. 5c) to an extend that it is now significant. We have, thus, revised Results, section 3.4., accordingly to discuss this new result and also looked into the individual lines within our mutant FUS pool (revised Fig. S8 and new supplemental Table S2). We found that the revised CC3 results are now in better agreement with our previously published CC3 data on the same lines (PMID: 29362359, Fig. 2j, k, l herein). In essence, we consistently observe augmented apoptosis in the untreated FUS-GFP P525L mutant in both studies at later time points, but not in the FUS 495QfsX527 mutant. Therefore, we wish to thank the reviewer for her/his help to reveal this reassuring consistency.

Round 3

Reviewer 1 Report

3.3 Rescue of axonal regeneration defects in FUS-ALS through MS.  

Major points

Response: The images of question in Fig. 4b show indeed the re-population of the distal assay area that was left blank right after the axotomy. The perceived density of the new axons here is a bit subjective but we agree with the reviewer that, due to the relatively subtle reduction of the growth cone speed in the untreated FUS mutant, it is hardly possible to spot any difference after just 24 hrs. Of note, however, is that we did not systematically investigated this in the presented setup since we had to select for traceable growth cones which might have biased such an analysis. We make this point now at the end of Results, section 3.3. Moreover, we revised the mid part of section 3.3. and M&M, section 2.11, and the discussion once more extensively to emphasize even more that (i) axons have progressively sprouted out from the distal channel exits in MFCs at 48 DIV before the axotomy was performed in all lines, (ii) that our axotomy created an entirely blank area to assay specifically the subsequent regenerative re-outgrowth (as opposed to general outgrowth in the course of regular differentiation) and (iii) that our mechanical truncation method is valid also considering several existing publications. Moreover, we have added a new Fig. S7 to the supplement to document the success of our complete axotomy with images from before and after the procedure.

From the images shown in Figure S7, it is impossible to appreciate what the authors subscribe to in the sentences highlighted in red.

1) After the axotomy, it is possible to appreciate the axons. Not all axons have been cut

2) The acquisition focus of the images is different between before and after treatment (axotomy). Note the microgrooves acquired on different levels.

NB There are several published papers that have shown faster growth after axotomy in ALS iPSC-derived MNs compared to its isogenic control line.

Response: We appreciate the reviewer’s point that we have to discuss our data more in light of conflicting other reports. We found that particularly Garone et al. 2021 (https://doi.org/10.1038/s42003-021-02538-8) found the opposite in what looks like a similar ALS-FUS model system (i.e. hiPSC-derived spinal MNs with a P525L mutation). We have added a new paragraph to our revised discussion to illuminate on possible reasons for the drastic, opposing discrepancy between our and their results. In essence, the reasons remain elusive but some feasible explanations are possible: different genetic backgrounds of the donors, different gene-editing technologies, different FUS-GFP-linkers, and, most importantly, different developmental stages when the axotomy was performed. Clearly, future studies are necessary for clarification.

Other papers besides Garone et al., 2021 suggest different results than those shown by the authors in this manuscript. Furthermore, Garone et al. 2021, demonstrated the rapid axonal growth phenotype also in a mouse model.

  1. McWhorter, M. L., Monani, U. R., Burghes, A. H. M. & Beattie, C. E. Knockdown of the survival motor neuron (Smn) protein in zebrafish causes defects in motor axon outgrowth and pathfinding. J. Cell Biol. 162, 919–931 (2003). 
  2. Osking, Z. et al. ALS-linked SOD1 mutants enhance neurite outgrowth and branching in adult motor neurons. iScience 11, 294–304 (2019).
  3. Garone MG, Birsa N, Rosito M, et al. ALS-related FUS mutations alter axon growth in motoneurons and affect HuD/ELAVL4 and FMRP activity. Commun Biol. 2021;4(1):1025. Published 2021 Sep 1. doi:10.1038/s42003-021-02538-8. 
  4. Marshall, K.L., Rajbhandari, L., Venkatesan, A. et al. Enhanced axonal regeneration of ALS patient iPSC-derived motor neurons harboring SOD1A4V mutation. Sci Rep 13, 5597 (2023). https://doi.org/10.1038/s41598-023-31720-7
  5.  

In the revised discussion, the authors cite a paper (Groen et al., 2013) to support their thesis, but this paper does not show axotomy analyses but rather axonal deficits. On this point, some documents prove precisely the opposite.

  1. Akiyama, T. et al. Aberrant axon branching via Fos-B dysregulation in FUSALS motor neurons. EBioMedicine 45, 362–378 (2019). 
  2. Osking, Z. et al. ALS-linked SOD1 mutants enhance neurite outgrowth and branching in adult motor neurons. iScience 11, 294–304 (2019).

Author Response

From the images shown in Figure S7, it is impossible to appreciate what the authors subscribe to in the sentences highlighted in red.

1) After the axotomy, it is possible to appreciate the axons. Not all axons have been cut

Response: In FUS-GFP WT there is a single neuron remaining in the lower channel and some cellular debris in the upper channel. The chances that this neuron is indeed undamaged by the axotomy is extremely low when examining the damage done to the rest of the neuronal network. It is very likely that this axon was indeed also sheared, if not in the macro channel than very likely in the microchannel. But even in the unlikely case that this single remaining neurite is not damaged at all, we want to point out, that the amount of axons, which is not sheared directly from the exit, is extremely low. Therefore, when imaging many growth cones the amount of false positive (not sheared) axon will be extremely low as well and thus the likelihood that these affect the quantifications is very small.

2) The acquisition focus of the images is different between before and after treatment (axotomy). Note the microgrooves acquired on different levels.

Response: The focus was done with the autofocus function of the microscope. Since there are no neurons left to appreciate the autofocus focused on the microchannels instead. Since this setup is using brightfield imaging the field of depth is large enough to visualize remaining axons. As pointed out by the review for the single axon in the lower channel of FUS-GFP WT. An intact axon in the FUS-GFP P525L image would look as light as in the FUS-GFP WT image if it was in wrong focus field. Thus this also will not influence the data analysis.

NB There are several published papers that have shown faster growth after axotomy in ALS iPSC-derived MNs compared to its isogenic control line.

Response: We appreciate the reviewer’s point that we have to discuss our data more in light of conflicting other reports. We found that particularly Garone et al. 2021 (https://doi.org/10.1038/s42003-021-02538-8) found the opposite in what looks like a similar ALS-FUS model system (i.e. hiPSC-derived spinal MNs with a P525L mutation). We have added a new paragraph to our revised discussion to illuminate on possible reasons for the drastic, opposing discrepancy between our and their results. In essence, the reasons remain elusive but some feasible explanations are possible: different genetic backgrounds of the donors, different gene-editing technologies, different FUS-GFP-linkers, and, most importantly, different developmental stages when the axotomy was performed. Clearly, future studies are necessary for clarification.

Other papers besides Garone et al., 2021 suggest different results than those shown by the authors in this manuscript. Furthermore, Garone et al. 2021, demonstrated the rapid axonal growth phenotype also in a mouse model.

  1. McWhorter, M. L., Monani, U. R., Burghes, A. H. M. & Beattie, C. E. Knockdown of the survival motor neuron (Smn) protein in zebrafish causes defects in motor axon outgrowth and pathfinding. J. Cell Biol. 162, 919–931 (2003). 
  2. Osking, Z. et al. ALS-linked SOD1 mutants enhance neurite outgrowth and branching in adult motor neurons. iScience 11, 294–304 (2019).
  3. Garone MG, Birsa N, Rosito M, et al. ALS-related FUS mutations alter axon growth in motoneurons and affect HuD/ELAVL4 and FMRP activity. Commun Biol. 2021;4(1):1025. Published 2021 Sep 1. doi:10.1038/s42003-021-02538-8. 
  4. Marshall, K.L., Rajbhandari, L., Venkatesan, A. et al. Enhanced axonal regeneration of ALS patient iPSC-derived motor neurons harboring SOD1A4V mutation. Sci Rep 13, 5597 (2023). https://doi.org/10.1038/s41598-023-31720-7
  5.  

In the revised discussion, the authors cite a paper (Groen et al., 2013) to support their thesis, but this paper does not show axotomy analyses but rather axonal deficits. On this point, some documents prove precisely the opposite.

  1. Akiyama, T. et al. Aberrant axon branching via Fos-B dysregulation in FUSALS motor neurons. EBioMedicine 45, 362–378 (2019). 
  2. Osking, Z. et al. ALS-linked SOD1 mutants enhance neurite outgrowth and branching in adult motor neurons. iScience 11, 294–304 (2019).

Response: WE are thankful for the reviewer to also mention this important literature. As the reviewer pointed out there is no consensus in the literature yet about axonal growth phenotypes. We, howver, elaborate once more to deepen the discussion on these on the first view different results, but also tried to find explanations why these might have been different. We now included above mentioned important paper in the discussion. The large differences are e.g. (1) species investigated, (2) mutations investigated, (3) only some looked on axonal regeneration after axonotomy as we did and (4) some used different timepoints of analysis after axonotomy as well as (5) the time point of differentiation when the analysis was performed. Especially the latter is of importance since we showed (also recently in doi: 10.1038/s41467-017-02299-1) that there is no initial axonal phenotype but only appears at later stages of differentiation (>21DIV), in which we then also documented the reduced axonal regrowth after axonotomy. We hope that this is now more clear in the revised version of our manuscript.

Reviewer 3 Report

I would like to thank the authors for critically revising their manuscript and am now happy to recommend it for publication.

Author Response

Reviewer 3:

I would like to thank the authors for critically revising their manuscript and am now happy to recommend it for publication.

Response: We deeply thank the reviewer for accepting our manuscript for publication.

Round 4

Reviewer 1 Report

I thank the authors for their effort and time dedicated to reviewing the manuscript.

The common denominator of the discrepancy between the outgrowth highlighted by the authors and the literature is to be referred exclusively to the maturation times of the NMs. I don't think  (i) the different donor patients and, therefore, the different genetic backgrounds, (ii) the different gene editing technologies used, (iii) the different linker sequences between FUS and the GFP tag, are the main reason for the changes in MN evolution after axotomy. 

As underlined by the authors, what unites all the existing literature are the experiments carried out at a time prior to what the authors describe in their manuscript. Therefore, I agree with the authors that further studies are needed to understand how maturation time affects regeneration after axotomy and how this is related to ALS. This is what the authors should highlight.